# SVDQuant: Absorbing Outliers by Low-Rank Components for 4-Bit Diffusion Models

**Muyang Li**[1][*][‡]   **Yujun Lin**[1][*]   **Zhekai Zhang**[1][†]   **Tianle Cai**[4]   **Xiuyu Li**[5][‡]
**Junxian Guo**[1,6]   **Enze Xie**[2]   **Chenlin Meng**[7]   **Jun-Yan Zhu**[3]   **Song Han**[1,2]
[1]MIT   [2]NVIDIA   [3]CMU   [4]Princeton   [5]UC Berkeley   [6]SJTU   [7]Pika Labs
https://hanlab.mit.edu/projects/svdquant

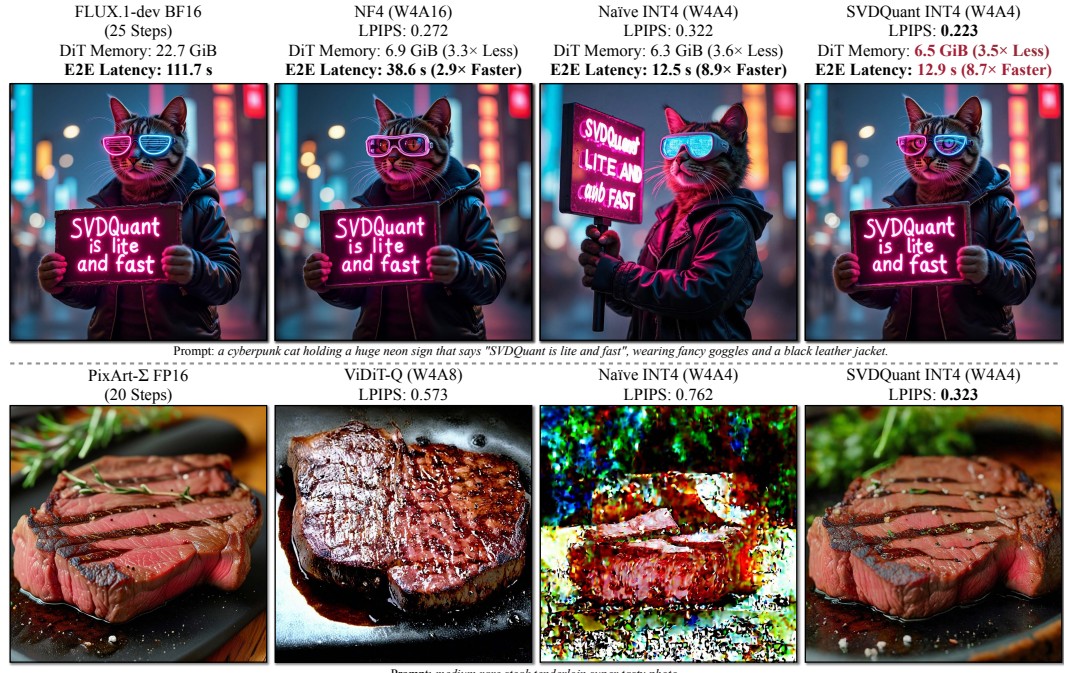

| FLUX.1-dev BF16 (25 Steps) DiT Memory: 22.7 GiB **E2E Latency: 111.7 s** | NF4 (W4A16) LPIPS: 0.272 DiT Memory: 6.9 GiB (3.3× Less) **E2E Latency: 38.6 s (2.9× Faster)** | Naïve INT4 (W4A4) LPIPS: 0.322 DiT Memory: 6.3 GiB (3.6× Less) **E2E Latency: 12.5 s (8.9× Faster)** | SVDQuant INT4 (W4A4) LPIPS: **0.223** DiT Memory: **6.5 GiB (3.5× Less)** **E2E Latency: 12.9 s (8.7× Faster)** |

Prompt: *a cyberpunk cat holding a huge neon sign that says "SVDQuant is lite and fast", wearing fancy goggles and a black leather jacket.*

| PixArt-Σ FP16 (20 Steps) | ViDiT-Q (W4A8) LPIPS: 0.573 | Naïve INT4 (W4A4) LPIPS: 0.762 | SVDQuant INT4 (W4A4) LPIPS: **0.323** |

Prompt: *medium rare steak tenderloin super tasty photo.*

Figure 1: SVDQuant is a post-training quantization technique for 4-bit weights and activations that well maintains visual fidelity. On 12B FLUX.1-dev, it achieves 3.6× memory reduction compared to the BF16 model. By eliminating CPU offloading, it offers 8.7× speedup over the 16-bit model when on a 16GB laptop 4090 GPU, 3× faster than the NF4 W4A16 baseline. On PixArt-Σ, it demonstrates significantly superior visual quality over other W4A4 or even W4A8 baselines. "E2E" means the end-to-end latency including the text encoder and VAE decoder.

## Abstract

Diffusion models can effectively generate high-quality images. However, as they scale, rising memory demands and higher latency pose substantial deployment challenges. In this work, we aim to accelerate diffusion models by quantizing their weights and activations to 4 bits. At such an aggressive level, both weights and activations are highly sensitive, where existing post-training quantization methods like smoothing become insufficient. To overcome this limitation, we propose *SVDQuant*, a new 4-bit quantization paradigm. Different from smoothing, which redistributes outliers between weights and activations, our approach *absorbs* these outliers using a low-rank branch. We first consolidate the outliers by shifting them from activations to weights. Then, we use a high-precision, low-rank branch to take in the weight outliers with Singular Value Decomposition (SVD), while a low-bit quantized branch handles the residuals. This process eases the quantization on both sides. However, naïvely running the low-rank branch independently incurs significant overhead due to extra data movement of activations, negating the quantization speedup. To address this, we co-design an inference engine *Nunchaku* that fuses the kernels of the low-rank branch into those of the low-bit branch to cut

---

[*]Algorithm co-lead. [†] System lead. [‡] Part of the work done during an internship at NVIDIA.

off redundant memory access. It can also seamlessly support off-the-shelf low-rank adapters (LoRAs) without re-quantization. Extensive experiments on SDXL, PixArt-Σ, and FLUX.1 validate the effectiveness of SVDQuant in preserving image quality. We reduce the memory usage for the 12B FLUX.1 models by 3.5×, achieving 3.0× speedup over the 4-bit weight-only quantization (W4A16) baseline on the 16GB laptop 4090 GPU with INT4 precision. On the latest RTX 5090 desktop with Blackwell architecture, we achieve a 3.1× speedup compared to the W4A16 model using NVFP4 precision. Our quantization library* and inference engine† are open-sourced.

# 1 INTRODUCTION

Diffusion models have shown remarkable capabilities in generating high-quality images (Ho et al., 2020), with recent advances further enhancing user control over the generation process. Trained on vast data, these models can create stunning images from simple text prompts, unlocking diverse image editing and synthesis applications (Meng et al., 2022b; Ruiz et al., 2023; Zhang et al., 2023).

To pursue higher image quality and more precise text-to-image alignment, researchers are scaling up diffusion models. As shown in Figure 2, Stable Diffusion (SD) (Rombach et al., 2022) 1.4 only has 800M parameters, while SDXL (Podell et al., 2024) scales this up to 2.6B parameters. AuraFlow v0.1 (fal.ai, 2024) extends this further to 6B parameters, with the latest model, FLUX.1 (Black-Forest-Labs, 2024), pushing the boundary to 12B parameters. Compared to large language models (LLMs), diffusion models are significantly more computationally intensive. Their computational costs‡ increase more rapidly with model size, posing a prohibitive memory and latency barrier for real-world model deployment, particularly for interactive use cases that demand low latency.

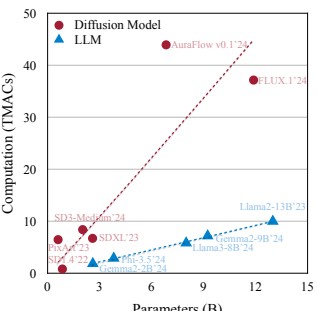

Figure 2: Computation *vs.* parameters for LLMs and diffusion models. LLMs' computation is measured with 512 context and 256 output tokens, and diffusion models' computation is for a single step. Dashed lines show trends.

As Moore's law slows down, hardware vendors are turning to low-precision inference to sustain performance improvements. For instance, NVIDIA's Blackwell Tensor Cores introduce a new 4-bit floating point (FP4) precision, doubling the performance compared to FP8 (NVIDIA, 2024). Therefore, using 4-bit inference to accelerate diffusion models is appealing. In the realm of LLMs, researchers have leveraged quantization to compress model sizes and boost inference speed (Dettmers et al., 2022; Xiao et al., 2023). However, unlike LLMs–where latency is primarily constrained by loading model weights on modern GPUs, especially with small batch sizes–diffusion models are heavily computationally bounded, even with a single batch. As a result, weight-only quantization cannot accelerate diffusion models. To achieve speedup on these devices, both weights and activations must be quantized to the same bit width; otherwise, the lower-precision weight will be upcast during computation, negating potential performance enhancements.

In this work, we focus on quantizing both the weights and activations of diffusion models to 4 bits. This challenging and aggressive scheme is often prone to severe quality degradation. Existing methods like smoothing (Xiao et al., 2023; Lin et al., 2024), which transfer the outliers between the weights and activations, are less effective since both sides are highly vulnerable to outliers. To address this issue, we propose a general-purpose quantization paradigm, *SVDQuant*. Our core idea is to use a low-cost branch to absorb outliers on both sides. To achieve this, as illustrated in Figure 3, we first aggregate the outliers by migrating them from activation $X$ to weight $W$ via smoothing. Then we apply Singular Value Decomposition (SVD) to the updated weight, $\hat{W}$, decomposing it into a low-rank branch $L_1 L_2$ and a residual $\hat{W} - L_1 L_2$. The low-rank branch operates at 16 bits, allowing us to quantize only the residual to 4 bits, significantly reducing outliers and magnitude. However, naively running the low-rank branch separately incurs substantial memory access overhead,

---

*Quantization library: github.com/mit-han-lab/deepcompressor
†Inference Engine: github.com/mit-han-lab/nunchaku
‡Measured by the number of Multiply-Accumulate operations (MACs). 1 MAC=2 FLOPs.

Figure 3: Overview of SVDQuant. (a) Originally, both the activation $X$ and weight $W$ contain outliers, making 4-bit quantization challenging. (b) We migrate the outliers from the activation to weight, resulting in the updated activation $\hat{X}$ and weight $\hat{W}$. While $\hat{X}$ becomes easier to quantize, $\hat{W}$ now becomes more difficult. (c) SVDQuant further decomposes $\hat{W}$ into a low-rank component $L_1 L_2$ and a residual $\hat{W} - L_1 L_2$ with SVD. Thus, the quantization difficulty is alleviated by the low-rank branch, which runs at 16-bit precision.

offsetting the speedup of 4-bit inference. To overcome this, we co-design a specialized inference engine *Nunchaku*, which fuses the low-rank branch computation into the 4-bit quantization and computation kernels. This design enables us to achieve measured inference speedup even with additional branches.

SVDQuant can quantize various text-to-image diffusion architectures into 4 bits, including both UNet (Ho et al., 2020; Ronneberger et al., 2015) and DiT (Peebles & Xie, 2023) backbones, while maintaining visual quality. It supports both INT4 and FP4 data types and integrates seamlessly with pre-trained low-rank adapters (LoRA) (Hsu et al., 2022) without requiring re-quantization. To our knowledge, we are the first to successfully apply 4-bit post-training quantization to both the weights and activations of diffusion models, and achieve measured speedup on NVIDIA GPUs. On the latest 12B FLUX.1, our 4-bit models largely preserve the image quality and reduce the memory footprint of the original BF16 model by 3.5×. Furthermore, our INT4 and FP4 model delivers a 3.0× and 3.1× speedup over the NF4 weight-only quantized baseline on the 16GB laptop-level RTX 4090 and desktop-level RTX 5090 GPU, respectively. See Figure 1 for visual examples.

## 2 RELATED WORK

**Diffusion models** (Sohl-Dickstein et al., 2015; Ho et al., 2020) have emerged as a powerful class of generative models, known for generating high-quality samples by modeling the data distribution through an iterative denoising process. Recent advancements in text-to-image diffusion models (Balaji et al., 2022; Rombach et al., 2022; Podell et al., 2024) have already revolutionized content generation. Researchers further shifted from convolution-based UNet architectures (Ronneberger et al., 2015; Ho et al., 2020) to transformers (Peebles & Xie, 2023; Bao et al., 2023) and scaled up the model size (Esser et al., 2024). However, diffusion models suffer from extremely slow inference speed due to their long denoising sequences and intense computation. To address this, various approaches have been proposed, including few-step samplers (Zhang & Chen, 2022; Zhang et al., 2022; Lu et al., 2022) or distilling fewer-step models from pre-trained ones (Salimans & Ho, 2021; Meng et al., 2022a; Song et al., 2023; Luo et al., 2023; Sauer et al., 2024; Yin et al., 2024b;a; kan, 2024). Another line of works choose to optimize or accelerate computation via efficient architecture design (Li et al., 2023b; 2020; Cai et al., 2024; Liu et al., 2024a), quantization (Shang et al., 2023; Li et al., 2023a), sparse inference (Li et al., 2022; Ma et al., 2024c;b), and distributed inference (Li et al., 2024b; Wang et al., 2024c; Chen et al., 2024b). This work focuses on quantizing the diffusion models to 4 bits to reduce the computation complexity. Our method can also be applied to few-step diffusion models to further reduce the latency (see Section 5.2).

**Quantization** has been recognized as an effective approach for LLMs to reduce the model size and accelerate inference (Dettmers et al., 2022; Frantar et al., 2023; Xiao et al., 2023; Lin et al., 2025; 2024; Kim et al., 2024; Zhao et al., 2024d). For diffusion models, Q-Diffusion (Li et al., 2023a) and PTQ4DM (Shang et al., 2023) first achieved 8-bit quantization. Subsequent works refined these techniques with approaches like sensitivity analysis (Yang et al., 2023) and timestep-aware quantization (He et al., 2023; Huang et al., 2024; Liu et al., 2024b; Wang et al., 2024a). Some recent works extended the setting to text-to-image models (Tang et al., 2024; Zhao et al., 2024c), DiT backbones (Wu et al., 2024), quantization-aware training (He et al., 2024; Zheng et al., 2024; Wang et al., 2024b; Sui et al., 2024), video generation (Zhao et al., 2024b), and different data types (Liu & Zhang, 2024). Among these works, only MixDQ (Zhao et al., 2024c) and ViDiT-Q (Zhao et al., 2024b) implement low-bit inference engines and report measured 8-bit speedup on GPUs. In this

work, we push the boundary further by quantizing diffusion models to 4 bits, supporting both the integer or floating-point data types, compatible with the UNet backbone (Ho et al., 2020) and recent DiT architecture (Peebles & Xie, 2023). Our co-designed inference engine, Nunchaku, further ensures on-hardware speedup. Additionally, when applying LoRA to the model, existing methods require fusing the LoRA branch to the main branch and re-quantizing the model to avoid tremendous memory-access overhead in the LoRA branch. Nunchaku cuts off this overhead via kernel fusion, allowing the low-rank branch to run efficiently as a separate branch, eliminating the need for re-quantization.

**Low-rank decomposition** has gained significant attention in deep learning for enhancing computational and memory efficiency (Hu et al., 2022; Zhao et al., 2024a; Jaiswal et al., 2024). While directly applying this approach to model weights can reduce the compute and memory demands (Hsu et al., 2022; Yuan et al., 2023; Li et al., 2023c), it often leads to performance degradation. Instead, Yao et al. (2024) combined it with quantization for model compression, employing a low-rank branch to compensate for the quantization error. Low-Rank Adaptation (LoRA) (Hu et al., 2022) introduces another active line of research using low-rank matrices to adjust a subset of pre-trained weights for efficient fine-tuning. This has sparked numerous advancements (Dettmers et al., 2023; Guo et al., 2024; Li et al., 2024c; Xu et al., 2024b; Meng et al., 2024), which combines quantized models with low-rank adapters to reduce memory usage during model fine-tuning. However, our work differs in two major aspects. First, our goal is different, as we aim to accelerate model inference through quantization, while previous works focus on model compression or efficient fine-tuning. Thus, they primarily consider weight-only quantization, resulting in no speedup. Second, as shown in our experiments (Figure 6 and ablation study in Section 5.2), directly applying these methods not only degrades the image quality, but also introduces significant overhead. In contrast, our method yields much better performance due to our joint quantization of weights and activations and overhead reduction of our inference engine Nunchaku.

## 3 QUANTIZATION PRELIMINARY

Quantization is an effective approach to accelerate linear layers in networks. Given a tensor $\boldsymbol{X}$, the quantization process is defined as:

$$\boldsymbol{Q_X} = \text{round}\left(\frac{\boldsymbol{X}}{s_{\boldsymbol{X}}}\right), s_{\boldsymbol{X}} = \frac{\max(|\boldsymbol{X}|)}{q_{\max}}. \tag{1}$$

Here, $\boldsymbol{Q_X}$ is the low-bit representation of $\boldsymbol{X}$, $s_{\boldsymbol{X}}$ is the scaling factor, and $q_{\max}$ is the maximum quantized value. For signed $k$-bit integer quantization, $q_{\max} = 2^{k-1} - 1$. For 4-bit floating-point quantization with 1-bit mantissa and 2-bit exponent, $q_{\max} = 6$. Thus, the dequantized tensor can be formulated as $Q(\boldsymbol{X}) = s_{\boldsymbol{X}} \cdot \boldsymbol{Q_X}$. For a linear layer with input $\boldsymbol{X}$ and weight $\boldsymbol{W}$, its computation can be approximated by

$$\boldsymbol{X}\boldsymbol{W} \approx Q(\boldsymbol{X})Q(\boldsymbol{W}) = s_{\boldsymbol{X}}s_{\boldsymbol{W}} \cdot \boldsymbol{Q_X}\boldsymbol{Q_W}. \tag{2}$$

The same approximation applies to convolutional layers. To speed up computation, modern commercial GPUs require both $\boldsymbol{Q_X}$ and $\boldsymbol{Q_W}$ using the same bit width. Otherwise, the low-bit weights need to be upcast to match the higher bit width of activations, or vice versa, negating the speed advantage. Following the notation in QServe (Lin et al., 2025), we denote $x$-bit weight, $y$-bit activation as W$x$A$y$. "INT" and "FP" refer to the integer and floating-point data types, respectively.

In this work, we focus on W4A4 quantization for acceleration, where outliers in both weights and activations place substantial obstacles. Traditional methods to suppress these outliers include quantization-aware training (QAT) (He et al., 2024) and rotation (Ashkboos et al., 2024; Liu et al., 2024c; Lin et al., 2025). QAT requires massive computing resources, especially for tuning models with more than 10B parameters such as FLUX.1. Rotation is inapplicable due to the usage of adaptive normalization layers (Peebles & Xie, 2023) in diffusion models. The runtime-generated normalization weights preclude the offline rotation with the weights of projection layers, while online rotation of both activations and weights incurs significant runtime overhead.

## 4 METHOD

In this section, we first formulate our problem and discuss where the quantization error comes from. Next, we present SVDQuant, a new W4A4 quantization paradigm for diffusion models. Our key

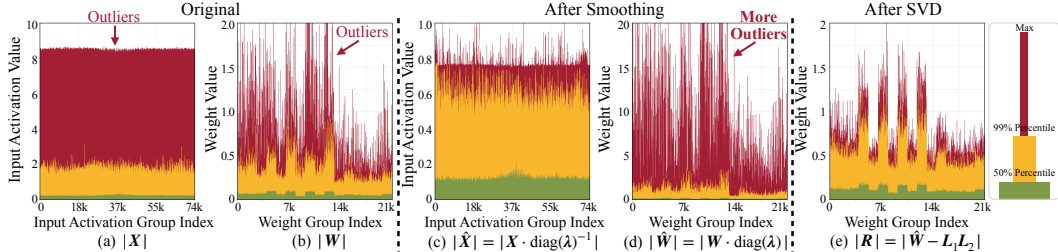

Figure 4: Example value distribution of inputs and weights in PixArt-$\Sigma$ (Chen et al., 2024a) $\boldsymbol{\lambda}$ is the smooth factor. Red indicates the outliers. Initially, both the input $\boldsymbol{X}$ and weight $\boldsymbol{W}$ contain significant outliers. After smoothing, the range of $\hat{\boldsymbol{X}}$ is reduced with much fewer outliers, while $\hat{\boldsymbol{W}}$ shows more outliers. Once the SVD low-rank branch $\boldsymbol{L}_1\boldsymbol{L}_2$ is subtracted, the residual $\boldsymbol{R}$ has a narrower range and is free from outliers.

idea is to introduce an additional low-rank branch that can absorb quantization difficulties in both weights and activations. Finally, we provide a co-designed inference engine Nunchaku with kernel fusion to minimize the overhead of the low-rank branches in the 4-bit model.

## 4.1 PROBLEM FORMULATION

Consider a linear layer with input $\boldsymbol{X} \in \mathbb{R}^{b \times m}$ and weight $\boldsymbol{W} \in \mathbb{R}^{m \times n}$, where $b$ represents the batch size, and $m$ and $n$ denote the input and output channels, respectively. The quantization error can be defined as

$$E(\boldsymbol{X}, \boldsymbol{W}) = \|\boldsymbol{X}\boldsymbol{W} - Q(\boldsymbol{X})Q(\boldsymbol{W})\|_F, \qquad (3)$$

where $\|\cdot\|_F$ denotes Frobenius Norm.

**Proposition 4.1** (Error decomposition). *The quantization error can be decomposed as follows:*

$$E(\boldsymbol{X}, \boldsymbol{W}) \leq \|\boldsymbol{X}\|_F \|\boldsymbol{W} - Q(\boldsymbol{W})\|_F + \|\boldsymbol{X} - Q(\boldsymbol{X})\|_F (\|\boldsymbol{W}\|_F + \|\boldsymbol{W} - Q(\boldsymbol{W})\|_F). \qquad (4)$$

See Appendix A.1 for the proof. From the proposition, we can see that the error is bounded by four elements – the magnitude of the weight and input, $\|\boldsymbol{W}\|_F$ and $\|\boldsymbol{X}\|_F$, and their respective quantization errors, $\|\boldsymbol{W} - Q(\boldsymbol{W})\|_F$ and $\|\boldsymbol{X} - Q(\boldsymbol{X})\|_F$. To minimize the overall quantization error, we aim to optimize these four terms.

## 4.2 SVDQUANT: ABSORBING OUTLIERS VIA LOW-RANK BRANCH

**Migrate outliers from activation to weight.** Smoothing (Xiao et al., 2023; Lin et al., 2024) is an effective approach for reducing outliers. We can smooth outliers in activations by scaling down the input $\boldsymbol{X}$ and adjusting the weight matrix $\boldsymbol{W}$ correspondingly using a per-channel smoothing factor $\boldsymbol{\lambda} \in \mathbb{R}^m$. As shown in Figure 4(a)(c), the smoothed input $\hat{\boldsymbol{X}} = \boldsymbol{X} \cdot \text{diag}(\boldsymbol{\lambda})^{-1}$ exhibits reduced magnitude and fewer outliers, resulting in lower input quantization error. However, in Figure 4(b)(d), the transformed weight $\hat{\boldsymbol{W}} = \boldsymbol{W} \cdot \text{diag}(\boldsymbol{\lambda})$ has a significant increase in both magnitude and the presence of outliers, which in turn raises the weight quantization error. Consequently, the overall error reduction is limited.

**Absorb magnified weight outliers with a low-rank branch.** Our core insight is to introduce a 16-bit low-rank branch to further migrate the weight quantization difficulty. Specifically, we decompose the transformed weight as $\hat{\boldsymbol{W}} = \boldsymbol{L}_1\boldsymbol{L}_2 + \boldsymbol{R}$, where $\boldsymbol{L}_1 \in \mathbb{R}^{m \times r}$ and $\boldsymbol{L}_2 \in \mathbb{R}^{r \times n}$ are two low-rank factors of rank $r$, and $\boldsymbol{R}$ is the residual. Then $\boldsymbol{X}\boldsymbol{W}$ can be approximated as

$$\boldsymbol{X}\boldsymbol{W} = \hat{\boldsymbol{X}}\hat{\boldsymbol{W}} = \hat{\boldsymbol{X}}\boldsymbol{L}_1\boldsymbol{L}_2 + \hat{\boldsymbol{X}}\boldsymbol{R} \approx \underbrace{\hat{\boldsymbol{X}}\boldsymbol{L}_1\boldsymbol{L}_2}_{\text{16-bit low-rank branch}} + \underbrace{Q(\hat{\boldsymbol{X}})Q(\boldsymbol{R})}_{\text{4-bit residual}}. \qquad (5)$$

Compared to direct 4-bit quantization, i.e., $Q(\hat{\boldsymbol{X}})Q(\boldsymbol{W})$, our method first computes the low-rank branch $\hat{\boldsymbol{X}}\boldsymbol{L}_1\boldsymbol{L}_2$ in 16-bit precision, and then approximates the residual $\hat{\boldsymbol{X}}\boldsymbol{R}$ with 4-bit quantization. Empirically, $r \ll \min(m, n)$, and is typically set to 16 or 32. As a result, the additional parameters and computation for the low-rank branch are negligible, contributing only $\frac{mr + nr}{mn}$ to the overall costs. However, it still requires careful system design to eliminate redundant memory access, which we will discuss in Section 4.3.

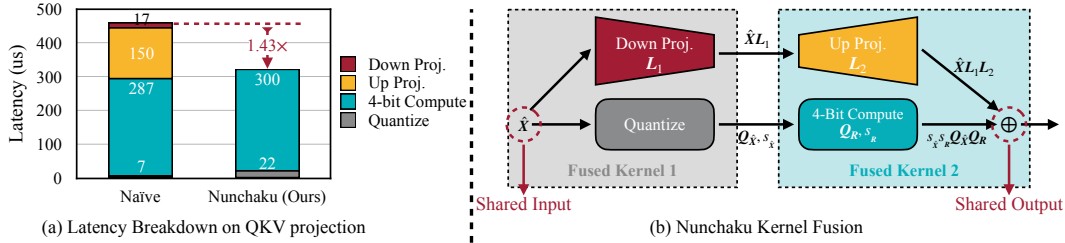

(a) Latency Breakdown on QKV projection        (b) Nunchaku Kernel Fusion

Figure 6: (a) Naïvely running low-rank branch with rank 32 will introduce 57% latency overhead due to extra read of 16-bit inputs in *Down Projection* and extra write of 16-bit outputs in *Up Projection*. Our Nunchaku engine optimizes this overhead with kernel fusion. (b) *Down Projection* and *Quantize* kernels use the same input, while *Up Projection* and *4-Bit Compute* kernels share the same output. To reduce data movement overhead, we fuse the first two and the latter two kernels together.

From Equation 5, the quantization error can be expressed as

$$\left\| \hat{X}\hat{W} - (\hat{X}L_1L_2 + Q(\hat{X})Q(R)) \right\|_F = \left\| \hat{X}R - Q(\hat{X})Q(R) \right\|_F = E(\hat{X}, R), \qquad (6)$$

where $R = \hat{W} - L_1L_2$. According to Proposition 4.1, since $\hat{X}$ is already free from outliers, we only need to focus on optimizing the magnitude of $R$, $\|R\|_F$ and its quantization error, $\|R - Q(R)\|_F$.

**Proposition 4.2** (Quantization error bound). *For any tensor $R$ and quantization method described in Equation 1 as $Q(R) = s_R \cdot Q_R$. Assuming the elements of $R$ follow a distribution that satisfies the following regularity condition: There exists a constant $c$ such that*

$$\mathbb{E}\left[\max(|R|)\right] \le c \cdot \mathbb{E}\left[\|R\|_F\right]. \qquad (7)$$

*Then, we have*

$$\mathbb{E}\left[\|R - Q(R)\|_F\right] \le \frac{c\sqrt{size(R)}}{q_{\max}} \cdot \mathbb{E}\left[\|R\|_F\right] \qquad (8)$$

*where $size(R)$ denotes the number of elements in $R$. Especially if the elements of $R$ follow a normal distribution, Equation 7 holds for $c = \sqrt{\frac{\log(size(R))\pi}{size(R)}}$.*

See Appendix A.2 for the proof. From this proposition, we obtain the intuition that the quantization error $\|R - Q(R)\|_F$ is bounded by the magnitude of the residual $\|R\|_F$. Thus, our goal is to find the optimal $L_1L_2$ that minimizes $\|R\|_F = \left\| \hat{W} - L_1L_2 \right\|_F$, which can be solved by Singular Value Decomposition (SVD) (Eckart & Young, 1936; Mirsky, 1960). Given the SVD of $\hat{W} = U\Sigma V$, the optimal solution is $L_1 = U\Sigma_{:,:r}$ and $L_2 = V_{:r,:}$. Figure 5 illustrates the singular value distribution of the original weight $W$, transformed weight $\hat{W}$ and residual $R$. The singular values of the original weight $W$ are highly imbalanced. After smoothing, the singular value distribution of $\hat{W}$ becomes even sharper, with only the first several values being significantly larger. By removing these dominant values, the magnitude of the residual $R$ is dramatically reduced, as $\|R\|_F =$

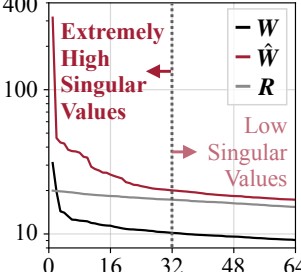

Figure 5: First 64 singular values of $W$, $\hat{W}$, and $R$. The first 32 singular values of $\hat{W}$ exhibit a steep drop, while the remaining values are much more gradual.

$\sqrt{\sum_{i=r+1}^{\min(m,n)} \sigma_i^2}$, compared to the original magnitude $\left\|\hat{W}\right\|_F = \sqrt{\sum_{i=1}^{\min(m,n)} \sigma_i^2}$, where $\sigma_i$ is the $i$-th singular value of $\hat{W}$. Furthermore, Figure 4(d)(e) show that $R$ exhibits fewer outliers with a substantially compressed value range compared to $\hat{W}$. In practice, we further reduce quantization errors by iteratively updating the low-rank branch through decomposing $W - Q(R)$ and adjusting $R$ accordingly for several iterations, and then picking the result with the smallest error.

### 4.3 NUNCHAKU: FUSING LOW-RANK AND LOW-BIT BRANCH KERNELS

Although the low-rank branch introduces negligible computation in theory, running it as a separate branch incurs large latency overhead—approximately 50% of the 4-bit branch latency, as shown in Figure 6(a). This occurs because, for a small rank $r$, even though the computational cost decreases

significantly, the input and output activation sizes remain unchanged, shifting the bottleneck from computation to memory access. This issue worsens, especially when the activation cannot fit into the GPU L2 cache. For example, in the diffusion transformer block, the up projection in the low-rank branch for QKV projection is much slower since its output exceeds the available L2 cache, resulting in the extra DRAM load and store operations. Fortunately, the down projection $L_1$ in the low-rank branch shares the same input as the quantization kernel in the low-bit branch, while the up projection $L_2$ shares the same output as the 4-bit computation kernel, as illustrated in Figure 6(b). By fusing the down projection with the quantization kernel and the up projection with the 4-bit computation kernel, the low-rank branch can share the activations with the low-bit branch, eliminating the extra memory access and halving the number of kernel calls. As a result, our low-rank branch adds only 5∼10% latency, making it nearly cost-free.

## 5 EXPERIMENTS

### 5.1 SETUPS

**Models.** We benchmark our methods using FLUX.1 (Black-Forest-Labs, 2024), PixArt-Σ (Chen et al., 2024a), SANA (Xie et al., 2025), Stable Diffusion XL (SDXL) (Podell et al., 2024) and SDXL-Turbo (Sauer et al., 2024), including both the UNet (Ronneberger et al., 2015; Ho et al., 2020) and DiT (Peebles & Xie, 2023) backbones. See Appendix B for more details.

**Datasets.** Following previous works (Li et al., 2023a; Zhao et al., 2024c;b), we randomly sample the prompts in COCO Captions 2024 (Chen et al., 2015) for calibration. To evaluate the generalization capability of our method, we sample 5K prompts from the MJHQ-30K (Li et al., 2024a) and the summarized Densely Captioned Images (sDCI) (Urbanek et al., 2024) for benchmarking. See Appendix C for more details.

**Baselines.** We compare SVDQuant against the following post-training quantization (PTQ) methods:

- 4-bit NormalFloat (NF4) is an information-theoretically optimal 4-bit data type for weight-only quantization (Dettmers et al., 2023), which assumes that weights follow a normal distribution. We use the community-quantized NF4 FLUX.1 models (Lllyasviel, 2024) as the baselines.
- ViDiT-Q (Zhao et al., 2024b) uses per-token quantization and smoothing (Xiao et al., 2023) to alleviate the outliers across different batches and tokens and achieves lossless 8-bit quantization on PixArt-Σ.
- MixDQ (Zhao et al., 2024c) identifies the outliers in the begin-of-sentence token of text embedding and protects them with 16-bit pre-computation. This method enables up to W4A8 quantization with negligible performance degradation on SDXL-Turbo.
- TensorRT contains an industry-level PTQ toolkit to quantize the diffusion models to 8 bits. It uses smoothing and only calibrates activations over a selected timestep range with a percentile scheme.

**Metrics.** Following previous works (Li et al., 2022; 2024b), we evaluate image quality and image similarity with respect to the 16-bit models' results. For image quality assessment, we use Fréchet Inception Distance (FID, lower is better) to measure the distribution distance between the generated images and the ground-truth images (Heusel et al., 2017; Parmar et al., 2022). Besides, we employ Image Reward (higher is better) to approximate the human rating of the generated images (Xu et al., 2024a). We use LPIPS (lower is better) to measure the perceptual similarity (Zhang et al., 2018) and Peak Signal Noise Ratio (PSNR, higher is better) to measure the numerical similarity of the images from the 16-bit models. Please refer to our Appendix E.1 for more metrics (CLIP IQA (Wang et al., 2023b), CLIP Score (Hessel et al., 2021) and SSIM[§]).

**Implementation details.** Please refer to Appendix D fore more details.

### 5.2 RESULTS

**Visual quality results.** We report the quantitative results in Table 1 across various models and precision levels, and show some corresponding 4-bit qualitative comparisons in Figure 7. Among all models, our 8-bit results can perfectly mirror the 16-bit results, achieving PSNR higher than 21, beating all other 8-bit baselines. On FLUX.1-dev, our INT8 PSNR even reaches 27 on MJHQ.

---

[§]https://en.wikipedia.org/wiki/Structural_similarity_index_measure

Table 1: Quantitative quality comparisons across different models. RTN stands for round-to-nearest. IR means ImageReward. Our 8-bit results closely match the quality of the 16-bit models. Moreover, our 4-bit results outperform other 4-bit baselines, effectively preserving the visual quality of 16-bit models.

| Backbone | Model | Precision | Method | MJHQ | | | | sDCI | | | |
|---|---|---|---|---|---|---|---|---|---|---|---|
| | | | | Quality | | Similarity | | Quality | | Similarity | |
| | | | | FID (↓) | IR (↑) | LPIPS (↓) | PSNR(↑) | FID (↓) | IR (↑) | LPIPS (↓) | PSNR (↑) |
| DiT | FLUX.1 -dev (50 Steps) | BF16 | – | 20.3 | 0.953 | – | – | 24.8 | 1.02 | – | – |
| | | INT W8A8 | Ours | 20.4 | 0.948 | 0.089 | 27.0 | 24.7 | 1.02 | 0.106 | 24.9 |
| | | W4A16 | NF4 | 20.6 | 0.910 | 0.272 | 19.5 | 24.9 | 0.986 | 0.292 | 18.2 |
| | | INT W4A4 | Ours | **19.9** | 0.935 | 0.223 | 21.0 | **24.2** | **1.01** | 0.240 | 19.7 |
| | | NVFP W4A4 | Ours | 20.4 | **0.937** | **0.208** | **21.4** | 24.7 | **1.01** | **0.218** | **20.2** |
| | FLUX.1 -schnell (4 Steps) | BF16 | – | 19.2 | 0.938 | – | – | 20.8 | 0.932 | – | – |
| | | INT W8A8 | Ours | 19.2 | 0.966 | 0.120 | 22.9 | 20.7 | 0.975 | 0.133 | 21.3 |
| | | W4A16 | NF4 | 18.9 | 0.943 | 0.257 | 18.2 | 20.7 | 0.953 | 0.263 | 17.1 |
| | | INT W4A4 | Ours | **18.3** | 0.951 | 0.258 | 18.3 | **20.1** | **0.979** | 0.260 | 17.2 |
| | | NVFP W4A4 | Ours | 19.0 | **0.968** | **0.227** | **19.0** | 20.5 | **0.979** | **0.226** | **18.1** |
| | PixArt-Σ (20 Steps) | FP16 | – | 16.6 | 0.944 | – | – | 24.8 | 0.966 | | |
| | | INT W8A8 | ViDiT-Q | **15.7** | 0.944 | 0.137 | 22.5 | **23.5** | **0.974** | 0.163 | 20.4 |
| | | INT W8A8 | Ours | 16.3 | **0.955** | **0.109** | **23.7** | 24.2 | 0.969 | **0.129** | **21.8** |
| | | INT W4A8 | ViDiT-Q | 37.3 | 0.573 | 0.611 | 12.0 | 40.6 | 0.600 | 0.629 | 11.2 |
| | | INT W4A4 | ViDiT-Q | 412 | -2.27 | 0.854 | 6.44 | 425 | -2.28 | 0.838 | 6.70 |
| | | INT W4A4 | Ours | 19.2 | 0.878 | 0.323 | 17.6 | 25.9 | 0.918 | 0.352 | 16.5 |
| | | NVFP W4A4 | Ours | **16.6** | **0.940** | **0.271** | **18.5** | **22.9** | **0.971** | **0.298** | **17.2** |
| | SANA -1.6B (20 Steps) | BF16 | – | 20.6 | 0.952 | – | – | 29.9 | 0.847 | – | – |
| | | INT W4A4 | RTN | 20.5 | 0.894 | 0.339 | 15.3 | 28.6 | 0.807 | 0.371 | 13.8 |
| | | INT W4A4 | Ours | **19.3** | 0.935 | 0.220 | 17.8 | **28.1** | 0.846 | 0.242 | 16.2 |
| | | NVFP W4A4 | RTN | 19.7 | 0.932 | 0.237 | 17.3 | 29.0 | 0.829 | 0.265 | 15.6 |
| | | NVFP W4A4 | Ours | 20.0 | **0.955** | **0.177** | **19.0** | 29.3 | **0.846** | **0.196** | **17.3** |
| UNet | SDXL -Turbo (4 Steps) | FP16 | – | 24.3 | 0.845 | – | – | 24.7 | 0.705 | – | – |
| | | INT W8A8 | MixDQ | **24.1** | 0.834 | 0.147 | 21.7 | 25.0 | 0.690 | 0.157 | 21.6 |
| | | INT W8A8 | Ours | 24.3 | **0.845** | **0.100** | **24.0** | **24.8** | **0.701** | **0.110** | **23.7** |
| | | INT W4A8 | MixDQ | 27.7 | 0.708 | 0.402 | 15.7 | 25.9 | 0.610 | 0.415 | 15.7 |
| | | INT W4A4 | MixDQ | 353 | -2.26 | 0.685 | 11.0 | 373 | -2.28 | 0.686 | 11.3 |
| | | INT W4A4 | Ours | 24.6 | 0.816 | 0.262 | 18.1 | 26.0 | 0.671 | 0.272 | 18.0 |
| | | NVFP W4A4 | Ours | **24.4** | **0.832** | **0.231** | **18.9** | **25.2** | **0.688** | **0.238** | **18.9** |
| | SDXL (30 Steps) | FP16 | – | 16.6 | 0.729 | – | – | 22.5 | 0.573 | – | – |
| | | INT W8A8 | TensorRT | 20.2 | 0.591 | 0.247 | 22.0 | 25.4 | 0.453 | 0.265 | 21.7 |
| | | INT W8A8 | Ours | **16.6** | **0.718** | **0.119** | **26.4** | **22.4** | **0.574** | **0.129** | **25.9** |
| | | INT W4A4 | Ours | 20.6 | 0.601 | 0.288 | 21.0 | 26.2 | 0.477 | 0.307 | 20.7 |
| | | NVFP W4A4 | Ours | **18.3** | **0.640** | **0.250** | **21.8** | **23.9** | **0.502** | **0.261** | **21.7** |

For 4-bit quantization, NVFP4 outperforms INT4, thanks to the native hardware support of smaller microscaling group size on Blackwell. On FLUX.1, our SVDQuant consistently surpasses the NF4 W4A16 baseline regarding all metrics. For the dev variant, our method even exceeds the original BF16 model regarding Image Reward, suggesting stronger human preference. On PixArt-Σ, while our INT4 method shows slight degradation, our NVFP4 model achieves a comparable score to the FP16 model. This is likely due to PixArt-Σ's highly compact model size (600M parameters), which benefits from a smaller group size. Remarkably, our INT4 and NVFP4 models significantly outperform ViDiT-Q's W4A8 results by a large margin across all metrics. Note that our FP16 PixArt-Σ model differs slightly from ViDiT's, though both offer the same quality. For fair comparisons, ViDiT-Q's similarity results are calculated using their FP16 results.

For UNet-based models, on SDXL-Turbo, our 4-bit models substantially outperform MixDQ W4A8, and our FID scores are on par with the FP16 models, indicating no quality loss. On SDXL, our INT4 and NVFP4 results achieve comparable quality to TensorRT's W8A8 performance, which represents the 8-bit SoTA. As shown in Figure 14 in the Appendix, our visual quality only shows minor degradation.

**Memory save and speedup.** In Figure 8, we report measured model size, memory savings, and speedup for FLUX.1. Our INT4 and NVFP4 quantization reduce the original transformer size from 22.2 GiB to 6.1 GiB, including a 0.3 GiB overhead due to the low-rank branch, resulting in an overall 3.6× reduction. Since both weights and activations are quantized, compared to the NF4

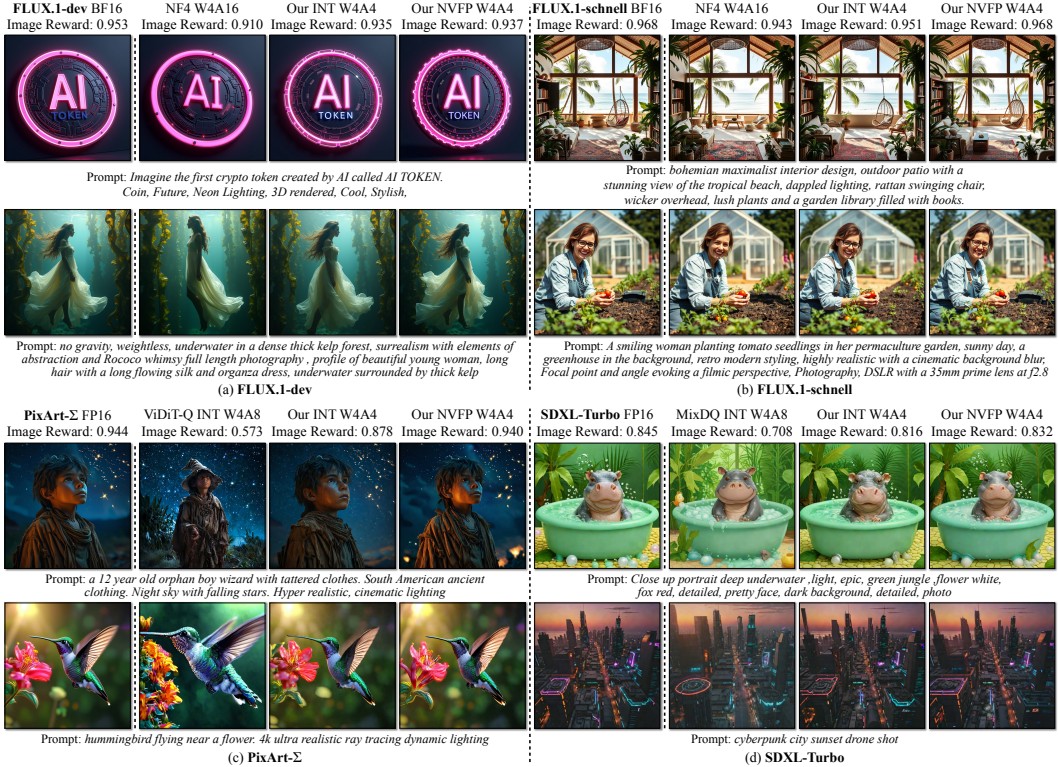

Figure 7: Qualitative visual results on MJHQ. Image Reward is calculated over the entire dataset. On FLUX.1 models, our 4-bit models outperform the NF4 W4A16 baselines, demonstrating superior text alignment and closer similarity to the 16-bit models. For instance, NF4 misses the swinging chair in the top right example. On PixArt-Σ and SDXL-Turbo, our 4-bit results demonstrate noticeably better visual quality than ViDiT-Q's and MixDQ's W4A8 results.

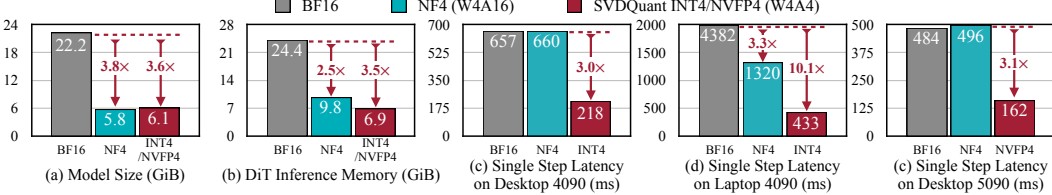

Figure 8: SVDQuant reduces the 12B FLUX.1 model size by 3.6× and cuts the 16-bit model's memory usage by 3.5×. With Nunchaku, our INT4 model runs 3.0× faster than the NF4 W4A16 baseline on both desktop and laptop NVIDIA RTX 4090 GPUs. Notably, on the laptop 4090, it achieves a total 10.1× speedup by eliminating CPU offloading. Our NVFP4 model is also 3.1× faster than both BF16 and NF4 on the RTX 5090 GPU.

weight-only-quantized variant, our inference engine Nunchaku even saves more memory footprint. It offers a 3.0× speedup on both desktop- and laptop-level NVIDIA RTX 4090 GPUs with INT4 precision and a 3.1× speedup on the RTX 5090 GPU with NVFP4 precision, compared to both NF4 and the original 16-bit models. Notably, while the original BF16 model requires per-layer CPU offloading on the 16GB laptop 4090, our INT4 model fits entirely in GPU memory, resulting in a 10.1× speedup by avoiding offloading.

**Integrate with LoRA.** Previous quantization methods require fusing the LoRA branches and re-quantizing the model when integrating LoRAs. In contrast, our Nunchaku eliminates redundant memory access, allowing adding a separate LoRA branch. In practice, we can fuse the LoRA branch into our low-rank branch by slightly increasing the rank, further enhancing efficiency. In Figure 9, we exhibit some visual examples of applying LoRAs of five different styles (Realism, Ghibsky Illustration, Anime, Children Sketch, and Yarn Art) to our INT4 FLUX.1-dev model. Our INT4 model successfully adapts to each style while preserving the image quality of the 16-bit version. For more visual examples, see Appendix E.2. For FLUX.1-schnell, we further support LoRAs from one-step conditional model pix2pix-turbo (Parmar et al., 2024), enabling additional controls like sketch. An interactive demo is available here.

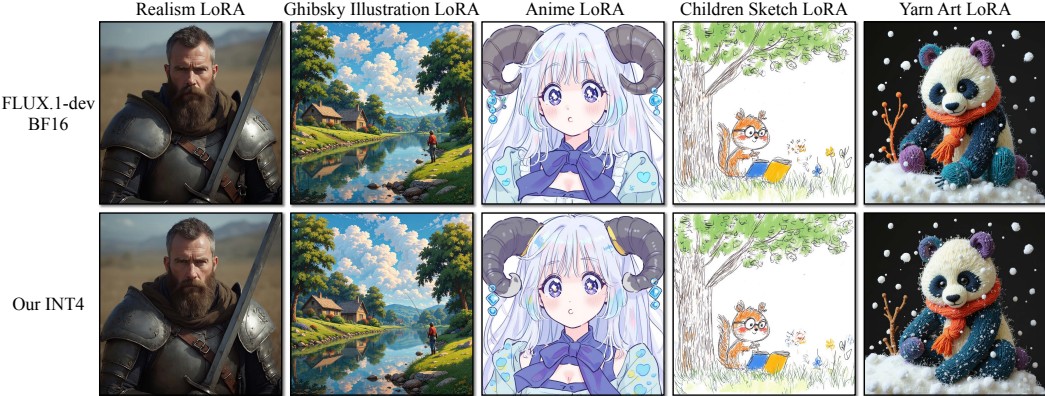

Figure 9: Our 4-bit model seamlessly integrates with off-the-shelf LoRAs without requiring requantization. When applying LoRAs, it matches the image quality of the original 16-bit FLUX.1-dev. See Appendix F for the text prompts.

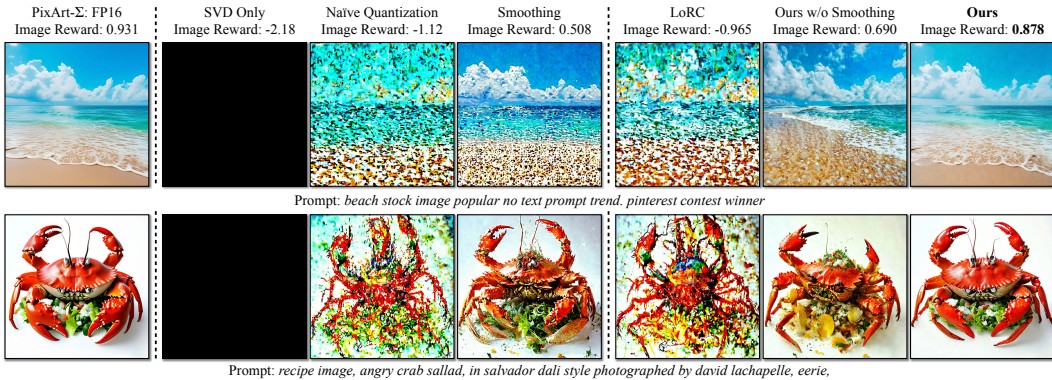

Figure 10: Ablation study of SVDQuant on PixArt-$\Sigma$. The rank of the low-rank branch is 64. Image Reward is measured over 1K samples from MJHQ. Our results significantly outperform the others, achieving the highest image quality by a wide margin.

**Ablation study.** In Figure 10, we present several ablation studies of SVDQuant on PixArt-$\Sigma$. First, both SVD-only and naïve quantization perform poorly in the 4-bit setting, resulting in severe quality degradation. While applying smoothing to the quantization slightly improves image quality compared to naïve quantization, the results remain unsatisfactory. LoRC (Yao et al., 2024) introduces a low-rank branch to compensate for quantization errors, but this approach is suboptimal, as quantization errors exhibit a well-spread distribution of singular values. Consequently, low-rank compensation fails to effectively mitigate these errors, as discussed in Section 4.2. In contrast, we first decompose the weights and quantize only the residual. As demonstrated in Figure 5, the first several singular values are significantly larger than the rest, allowing us to shift them to the low-rank branch, effectively reducing weight magnitude. Finally, smoothing consolidates the outliers, enabling the low-rank branch to absorb outliers from the activations and substantially improving image quality.

**Trade-off of increasing rank.** Please refer to Appendix E.5 for more details.

## 6 CONCLUSION

In this work, we introduce a novel 4-bit post-training quantization paradigm, SVDQuant, for diffusion models. It adopts a low-rank branch to absorb the outliers in both the weights and activations, easing the process of quantization. Our inference engine Nunchaku further fuses the low-rank and low-bit branch kernels, reducing memory usage and cutting off redundant data movement overhead. Extensive experiments demonstrate that SVDQuant preserves image quality. Nunchaku further achieves a 3.5× reduction in memory usage over the original 16-bit model and 3.0× speedup over the W4A16 on an NVIDIA RTX 4090 and 5090 GPUs. This advancement enables the efficient deployment of large-scale diffusion models on edge devices, unlocking broader potential for interactive AI applications.

## ACKNOWLEDGMENTS

We thank NVIDIA for donating the DGX server and Blackwell GPUs. We thank MIT-IBM Watson AI Lab, MIT and Amazon Science Hub, MIT AI Hardware Program, National Science Foundation, Sloan foundation, Packard Foundation, Dell, LG, Hyundai, and Samsung for supporting this research. We thank Paulius Micikevicius for his support and discussion.

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

## A PROOFS

### A.1 PROOF OF PROPOSITION 4.1

**Proposition 4.1.** *The quantization error $E(\boldsymbol{X}, \boldsymbol{W}) = \|\boldsymbol{X}\boldsymbol{W} - Q(\boldsymbol{X})Q(\boldsymbol{W})\|_F$ in Equation 3 can be decomposed as follows:*

$$E(\boldsymbol{X}, \boldsymbol{W}) \leq \|\boldsymbol{X}\|_F \|\boldsymbol{W} - Q(\boldsymbol{W})\|_F + \|\boldsymbol{X} - Q(\boldsymbol{X})\|_F (\|\boldsymbol{W}\|_F + \|\boldsymbol{W} - Q(\boldsymbol{W})\|_F). \quad (9)$$

*Proof.*

$$\begin{aligned}
&\|\boldsymbol{X}\boldsymbol{W} - Q(\boldsymbol{X})Q(\boldsymbol{W})\|_F \\
=& \|\boldsymbol{X}\boldsymbol{W} - \boldsymbol{X}Q(\boldsymbol{W}) + \boldsymbol{X}Q(\boldsymbol{W}) - Q(\boldsymbol{X})Q(\boldsymbol{W})\|_F \\
\leq& \|\boldsymbol{X}(\boldsymbol{W} - Q(\boldsymbol{W}))\|_F + \|(\boldsymbol{X} - Q(\boldsymbol{X}))Q(\boldsymbol{W})\|_F \\
\leq& \|\boldsymbol{X}\|_F \|\boldsymbol{W} - Q(\boldsymbol{W})\|_F + \|\boldsymbol{X} - Q(\boldsymbol{X})\|_F \|Q(\boldsymbol{W})\|_F \\
\leq& \|\boldsymbol{X}\|_F \|\boldsymbol{W} - Q(\boldsymbol{W})\|_F + \|\boldsymbol{X} - Q(\boldsymbol{X})\|_F \|\boldsymbol{W} - (\boldsymbol{W} - Q(\boldsymbol{W}))\|_F \\
\leq& \|\boldsymbol{X}\|_F \|\boldsymbol{W} - Q(\boldsymbol{W})\|_F + \|\boldsymbol{X} - Q(\boldsymbol{X})\|_F (\|\boldsymbol{W}\|_F + \|\boldsymbol{W} - Q(\boldsymbol{W})\|_F).
\end{aligned}$$

$\square$

### A.2 PROOF OF PROPOSITION 4.2

**Proposition 4.2.** *For any tensor $\boldsymbol{R}$ and quantization method described in Equation 1 as $Q(\boldsymbol{R}) = s_{\boldsymbol{R}} \cdot \boldsymbol{Q_R}$. Assuming the elements of $\boldsymbol{R}$ follow a distribution that satisfies the following regularity condition: There exists a constant $c$ such that*

$$\mathbb{E}\left[\max(|\boldsymbol{R}|)\right] \leq c \cdot \mathbb{E}\left[\|\boldsymbol{R}\|_F\right]. \quad (10)$$

*Then, we have*

$$\mathbb{E}\left[\|\boldsymbol{R} - Q(\boldsymbol{R})\|_F\right] \leq \frac{c\sqrt{size(\boldsymbol{R})}}{q_{\max}} \cdot \mathbb{E}\left[\|\boldsymbol{R}\|_F\right] \quad (11)$$

*where $size(\boldsymbol{R})$ denotes the number of elements in $\boldsymbol{R}$. Especially if the elements of $\boldsymbol{R}$ follow a normal distribution, Equation 10 holds for $c = \sqrt{\frac{\log(size(\boldsymbol{R}))\pi}{size(\boldsymbol{R})}}$.*

*Proof.*

$$\begin{aligned}
&\|\boldsymbol{R} - Q(\boldsymbol{R})\|_F \\
=& \|\boldsymbol{R} - s_{\boldsymbol{R}} \cdot \boldsymbol{Q_R}\|_F \\
=& \left\|s_{\boldsymbol{R}} \cdot \frac{\boldsymbol{R}}{s_{\boldsymbol{R}}} - s_{\boldsymbol{R}} \cdot \mathrm{round}\left(\frac{\boldsymbol{R}}{s_{\boldsymbol{R}}}\right)\right\|_F \\
=& |s_{\boldsymbol{R}}| \left\|\frac{\boldsymbol{R}}{s_{\boldsymbol{R}}} - \mathrm{round}\left(\frac{\boldsymbol{R}}{s_{\boldsymbol{R}}}\right)\right\|_F.
\end{aligned}$$

So,

$$\begin{aligned}
&\mathbb{E}\left[\|\boldsymbol{R} - Q(\boldsymbol{R})\|_F\right] \\
\leq& \mathbb{E}\left[|s_{\boldsymbol{R}}|\right]\sqrt{\mathrm{size}(\boldsymbol{R})} \\
=& \frac{\sqrt{\mathrm{size}(\boldsymbol{R})}}{q_{\max}} \cdot \mathbb{E}\left[\max(|\boldsymbol{R}|)\right] \\
\leq& \frac{c\sqrt{\mathrm{size}(\boldsymbol{R})}}{q_{\max}} \cdot \mathbb{E}\left[\|\boldsymbol{R}\|_F\right]
\end{aligned}$$

Especially, if the elements of $\boldsymbol{R}$ follows a normal distribution, we have

$$\mathbb{E}\left[\max(|\boldsymbol{R}|)\right] \leq \sigma\sqrt{2\log\left(\mathrm{size}(\boldsymbol{R})\right)} \quad (12)$$

where $\sigma$ is the std deviation of the normal distribution. Equation 12 comes from the maximal inequality of Gaussian variables (Lemma 2.3 in Massart (2007)).

On the other hand,

$$\mathbb{E}\left[\|\boldsymbol{R}\|_F\right]$$

$$=\mathbb{E}\left[\sqrt{\sum_{x\in\boldsymbol{R}}x^2}\right]$$

$$\geq\mathbb{E}\left[\frac{\sum_{x\in\boldsymbol{R}}|x|}{\sqrt{\mathrm{size}(\boldsymbol{R})}}\right] \tag{13}$$

$$=\sigma\sqrt{\frac{2\mathrm{size}(\boldsymbol{R})}{\pi}}, \tag{14}$$

where Equation 13 comes from Cauchy-Schwartz inequality and Equation 14 comes from the expectation of half-normal distribution.

Together, we have that for a normal distribution,

$$\mathbb{E}\left[\max(|\boldsymbol{R}|)\right]$$

$$\leq\sigma\sqrt{2\log\left(\mathrm{size}(\boldsymbol{R})\right)}$$

$$\leq\sqrt{\frac{\log\left(\mathrm{size}(\boldsymbol{R})\right)\pi}{\mathrm{size}(\boldsymbol{R})}}\mathbb{E}\left[\|\boldsymbol{R}\|_F\right].$$

In other words, Equation 10 holds for $c=\sqrt{\frac{\log(\mathrm{size}(\boldsymbol{R}))\pi}{\mathrm{size}(\boldsymbol{R})}}$. $\square$

## B  BENCHMARK MODELS

We benchmark our methods using the following six text-to-image models:

- FLUX.1 (Black-Forest-Labs, 2024) is the SoTA open-sourced DiT-based diffusion model. It consists of 19 joint attention blocks (Esser et al., 2024) and 38 parallel attention blocks (Dehghani et al., 2023), totaling 12B parameters. We evaluate both the 50-step guidance-distilled (FLUX.1-dev) and 4-step timestep-distilled (FLUX.1-schnell) variants.

- PixArt-$\Sigma$ (Chen et al., 2024a) is another DiT-based model. Instead of using joint attention, it stacks 28 attention blocks composed of self-attention, cross-attention, and feed-forward layers, amounting to 600M parameters. We evaluate it on the default 20-step setting.

- SANA (Xie et al., 2025) is a 1.6B DiT model. It utilizes a 32× compression autoencoder (Chen et al., 2025) and replaces Softmax attention with linear attention to accelerate image generation.

- Stable Diffusion XL (SDXL) is a widely-used UNet-based model with 2.6B parameters (Podell et al., 2024). It predicts noise with three resolution scales. The highest-resolution stage is processed entirely by ResBlocks (He et al., 2016), while the other two stages jointly use ResBlocks and attention layers. Like PixArt-$\Sigma$, SDXL uses cross-attention layers for text conditioning. We evaluate it in the 30-step setting, along with its 4-step distilled variant, SDXL-Turbo (Sauer et al., 2024).

## C  BENCHMARK DATASETS

To assess the generalization capability of our method, we adopt two distinct prompt sets with varying styles for benchmarking:

- MJHQ-30K (Li et al., 2024a) consists of 30K samples from Midjourney with 10 common categories, 3K samples each. We randomly select 5K prompts from this dataset to evaluate model performance on artistic image generation.

- Densely Captioned Images (DCI) (Urbanek et al., 2024) is a dataset containing $\sim$8K images with detailed human-annotated captions, averaging over 1,000 words. For our experiments, we use

the summarized version (sDCI), where captions are condensed to 77 tokens using large language models (LLMs) to accommodate diffusion models. Similarly, we randomly sample 5K prompts for realistic image generation.

# D    IMPLEMENTATION DETAILS

For the 8-bit setting, we use per-token dynamic activation quantization and per-channel weight quantization with a low-rank branch of rank 16. For the 4-bit setting, we adopt per-group symmetric quantization for both activations and weights, along with a low-rank branch of rank 32. INT4 quantization uses a group size of 64 with 16-bit scales. We use NVFP4 for FP4 quantization, which has native hardware support of group size of 16 with FP8 scales on Blackwell GPUs (NVIDIA Corporation, 2025). We use GPTQ (Frantar et al., 2023) to quantize the residual weights. For FLUX.1 models, the inputs of linear layers in adaptive normalization are kept in 16 bits (*i.e.*, W4A16). For other models, key and value projections in the cross-attention are retained at 16 bits since their latency only covers less than 5% of total runtime.

The smoothing factor $\lambda \in \mathbb{R}^m$ is a per-channel vector whose $i$-th element is computed as $\lambda_i = \max(|\boldsymbol{X}_{:,i}|)^\alpha / \max(|\boldsymbol{W}_{i,:}|)^{1-\alpha}$ following SmoothQuant (Xiao et al., 2023) Here, $\boldsymbol{X} \in \mathbb{R}^{b \times m}$ and $\boldsymbol{W} \in \mathbb{R}^{m \times n}$. It is decided offline by searching for the best migration strength $\alpha$ for each layer to minimize the layer output mean squared error (MSE) after SVD on the calibration dataset.

# E ADDITIONAL RESULTS

## E.1 VISUAL QUALITY RESULTS

We report extra quantitative quality results with additional metrics in Table 2. Specifically, CLIP IQA (Wang et al., 2023b) and CLIP Score (Hessel et al., 2021) assesses the image quality and text-image alignment with CLIP (Radford et al., 2021), respectively. Structural Similarity Index Measure (SSIM) is used to measure the luminance, contrast, and structure similarity of images produced by our 4-bit model against the original 16-bit model. We also visualize more qualitative comparsions in Figures 11, 12, 13, 14 and 15.

Table 2: Additional quantitative quality comparisons across different models. RTN stands for round-to-nearest. C.IQA means CLIP IQA, and C.SCR means CLIP Score.

| Backbone | Model | Precision | Method | MJHQ | | | sDCI | | |
|---|---|---|---|---|---|---|---|---|---|
| | | | | Quality | | Similarity | Quality | | Similarity |
| | | | | C.IQA (↑) | C.SCR (↑) | SSIM(↑) | C.IQA (↑) | C.SCR (↑) | SSIM (↑) |
| DiT | FLUX.1 -dev (50 Steps) | BF16 | – | 0.952 | 26.0 | – | 0.955 | 25.4 | – |
| | | INT W8A8 | Ours | 0.953 | 26.0 | 0.748 | 0.955 | 25.4 | 0.697 |
| | | W4A16 | NF4 | 0.947 | **25.8** | 0.748 | 0.951 | **25.4** | 0.697 |
| | | INT W4A4 | Ours | 0.950 | **25.8** | 0.797 | 0.951 | 25.3 | 0.751 |
| | | NVFP W4A4 | Ours | **0.952** | **25.8** | **0.808** | **0.955** | **25.4** | **0.768** |
| | FLUX.1 -schnell (4 Steps) | BF16 | – | 0.938 | 26.6 | – | 0.932 | 26.2 | – |
| | | INT W8A8 | Ours | 0.938 | 26.6 | 0.844 | 0.932 | 26.2 | 0.811 |
| | | W4A16 | NF4 | **0.941** | 26.6 | 0.713 | **0.933** | 26.2 | 0.674 |
| | | INT W4A4 | Ours | 0.937 | 26.5 | 0.720 | 0.932 | **26.2** | 0.681 |
| | | NVFP W4A4 | Ours | 0.939 | **26.6** | **0.745** | 0.932 | 26.1 | **0.712** |
| | PixArt-Σ (20 Steps) | FP16 | – | 0.944 | 26.8 | – | 0.966 | 26.1 | – |
| | | INT W8A8 | ViDiT-Q | **0.948** | 26.7 | 0.815 | 0.966 | **26.1** | 0.756 |
| | | INT W8A8 | Ours | 0.947 | **26.8** | **0.849** | **0.967** | 26.0 | **0.800** |
| | | INT W4A8 | ViDiT-Q | 0.912 | 25.7 | 0.356 | 0.917 | 25.4 | 0.295 |
| | | INT W4A4 | ViDiT-Q | 0.185 | 13.3 | 0.077 | 0.176 | 13.3 | 0.080 |
| | | INT W4A4 | Ours | 0.926 | 26.6 | 0.655 | 0.948 | **26.1** | 0.577 |
| | | NVFP W4A4 | Ours | **0.938** | **26.7** | **0.692** | **0.956** | **26.1** | **0.618** |
| | SANA -1.6B (20 Steps) | BF16 | – | 0.934 | 26.8 | – | 0.958 | 26.4 | – |
| | | INT W4A4 | RTN | 0.915 | **26.9** | 0.604 | 0.943 | **26.4** | 0.538 |
| | | INT W4A4 | Ours | 0.926 | **26.9** | 0.710 | 0.951 | **26.4** | 0.649 |
| | | NVFP W4A4 | RTN | 0.929 | 26.8 | 0.694 | 0.953 | **26.4** | 0.626 |
| | | NVFP W4A4 | Ours | **0.932** | **26.9** | **0.755** | **0.955** | **26.4** | **0.701** |
| UNet | SDXL -Turbo (4 Steps) | FP16 | – | 0.926 | 26.5 | – | 0.913 | 26.5 | – |
| | | INT W8A8 | MixDQ | 0.922 | 26.5 | 0.763 | 0.907 | 26.5 | 0.750 |
| | | INT W8A8 | Ours | **0.925** | 26.5 | **0.821** | **0.912** | 26.5 | **0.808** |
| | | INT W4A8 | MixDQ | 0.893 | 25.9 | 0.512 | **0.895** | 26.1 | 0.493 |
| | | INT W4A4 | MixDQ | 0.556 | 13.1 | 0.289 | 0.548 | 11.9 | 0.296 |
| | | INT W4A4 | Ours | 0.915 | 26.5 | 0.631 | 0.894 | 26.8 | 0.614 |
| | | FP W4A4 | Ours | **0.919** | 26.5 | **0.663** | 0.902 | 26.6 | **0.649** |
| | SDXL (30 Steps) | FP16 | – | 0.907 | 27.2 | – | 0.911 | 26.5 | – |
| | | INT W8A8 | TensorRT | 0.905 | 26.7 | 0.733 | 0.901 | 26.1 | 0.697 |
| | | INT W8A8 | Ours | **0.912** | **27.0** | **0.843** | **0.910** | **26.3** | **0.814** |
| | | INT W4A4 | Ours | 0.878 | 26.7 | 0.717 | 0.862 | 26.2 | 0.672 |
| | | NVFP W4A4 | Ours | **0.892** | **26.8** | **0.739** | **0.877** | **26.4** | **0.701** |

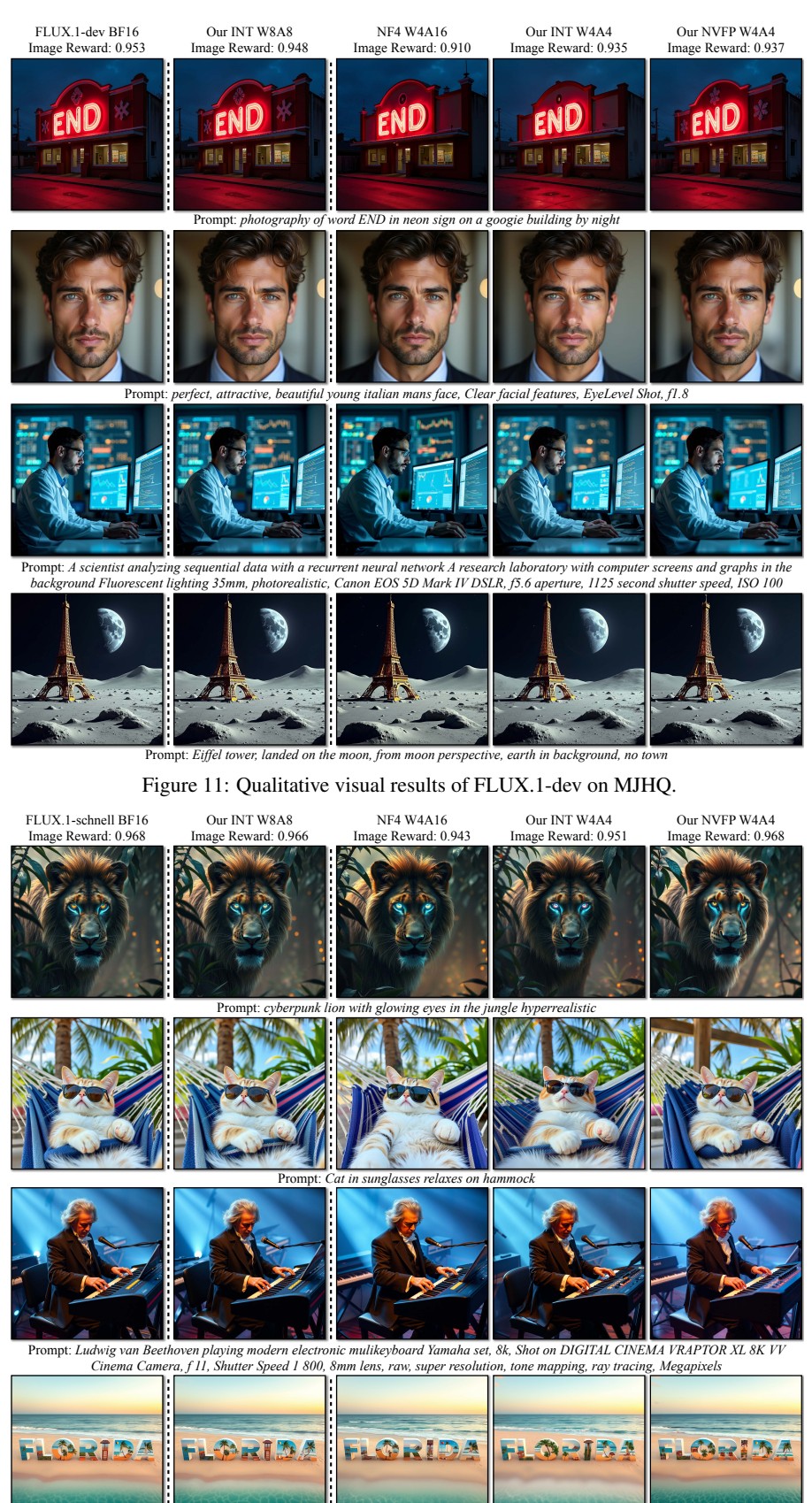

Figure 11: Qualitative visual results of FLUX.1-dev on MJHQ.

Figure 12: Qualitative visual results of FLUX.1-schnell on MJHQ.

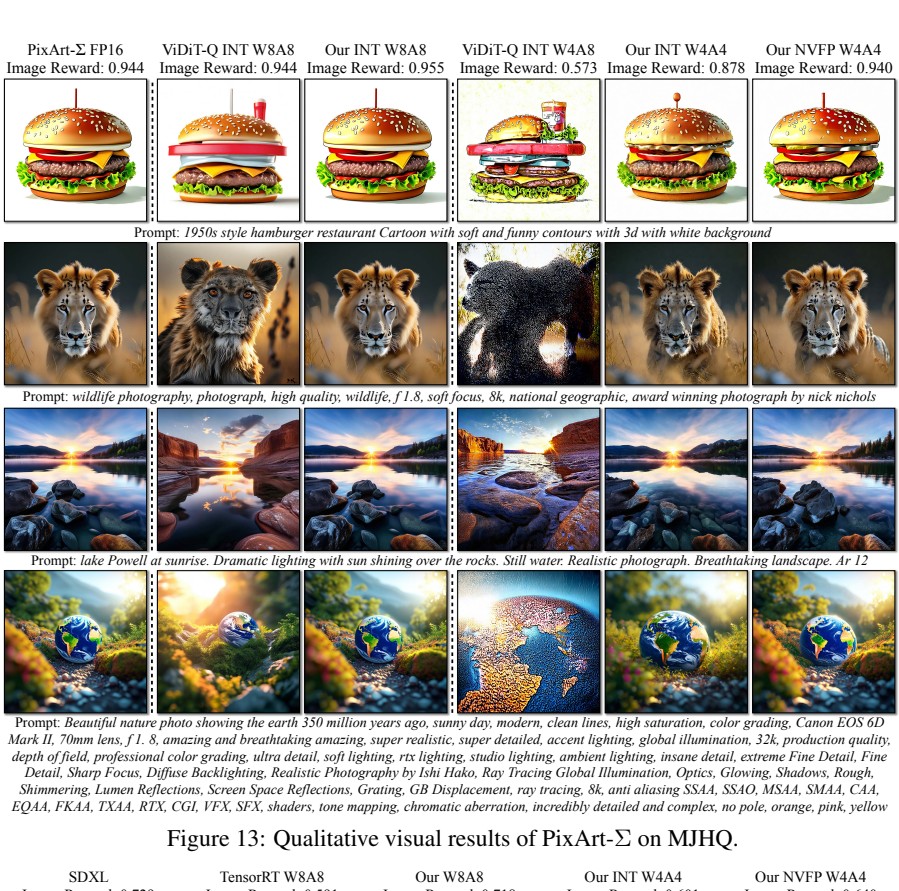

Figure 13: Qualitative visual results of PixArt-Σ on MJHQ.

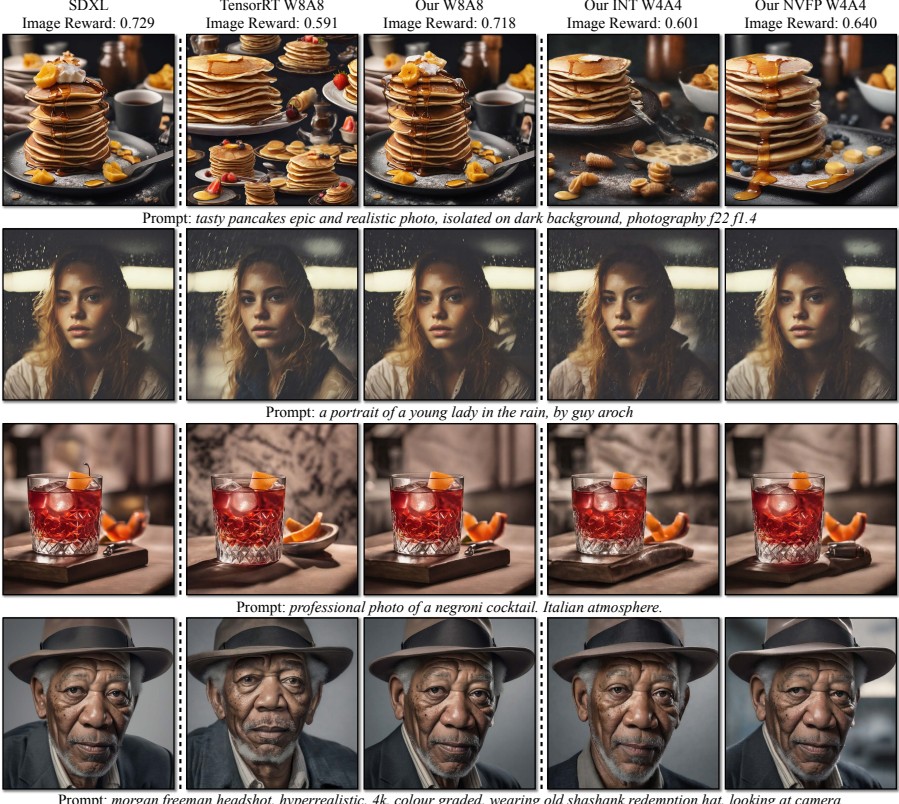

Figure 14: Qualitative visual results of SDXL on MJHQ.

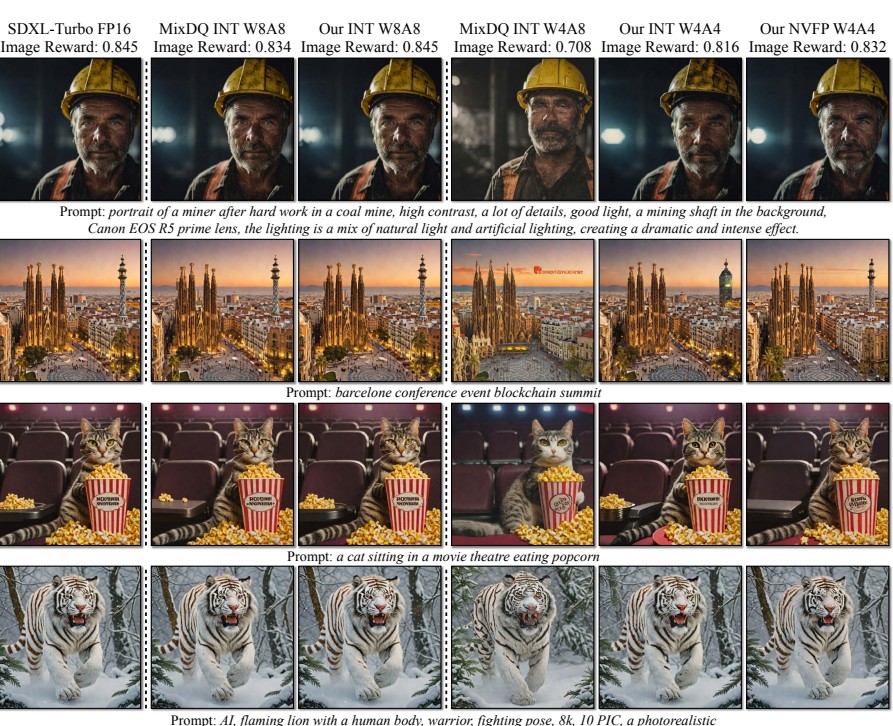

Figure 15: Qualitative visual results of SDXL-Turbo on MJHQ.

## E.2 LoRA RESULTS

In Figure 16, we showcase more visual results of applying the aforementioned five community-contributed LoRAs of different styles (Realism, Ghibsky Illustration, Anime, Children Sketch, and Yarn Art) to our INT4 quantized models.

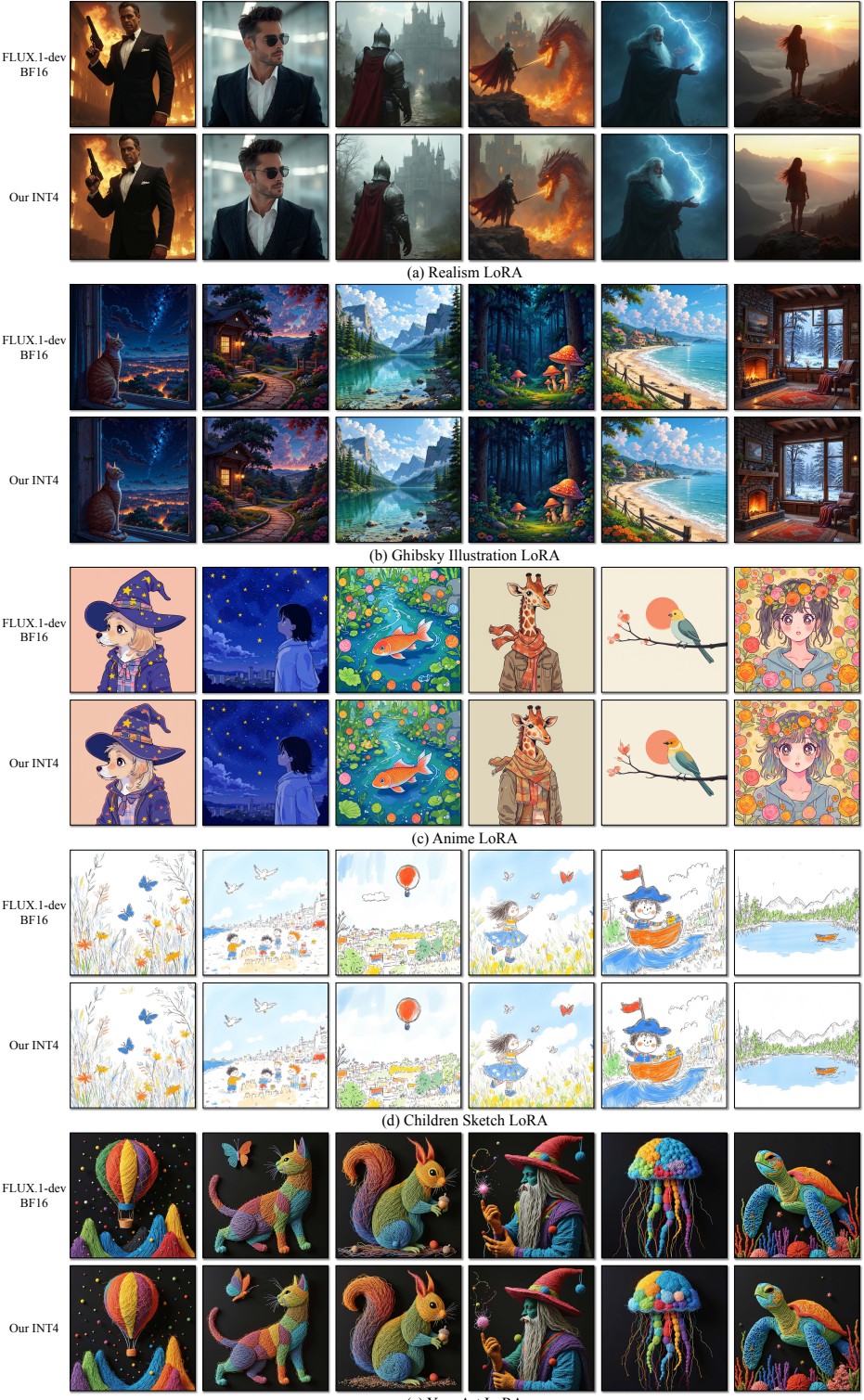

Figure 16: Additional LoRA results on FLUX.1-dev. When applying LoRAs, our INT4 model matches the image quality of the original BF16 model. See Appendix F for the detailed used text prompts.

### E.3 ADDITIONAL ABLATION OF SVDQUANT

Table 3: Quantitative comparisons of different SVDQuant settings on MJHQ. NVFP4 outperforms INT4. SVDQuant leverages a low-rank branch to ease quantization, significantly enhancing image quality. It can further apply GPTQ to quantize the weight residual, further improving quality.

| Model | Precision | Low-rank Branch | GPTQ | Image Reward (↑) | LPIPS (↓) | PSNR (↑) |
|---|---|---|---|---|---|---|
| | BF16 | – | – | 0.953 | – | – |
| | INT4 | ✗ | ✗ | 0.908 | 0.322 | 18.5 |
| | | ✗ | ✓ | 0.933 | 0.297 | 19.1 |
| | | ✓ | ✗ | 0.926 | 0.256 | 20.1 |
| FLUX.1-dev | | ✓ | ✓ | **0.935** | **0.223** | **21.0** |
| | NVFP4 | ✗ | ✗ | 0.928 | 0.244 | 20.3 |
| | | ✗ | ✓ | 0.936 | **0.204** | **21.5** |
| | | ✓ | ✗ | 0.935 | 0.223 | 20.9 |
| | | ✓ | ✓ | **0.937** | 0.208 | 21.4 |
| | BF16 | – | – | 0.968 | – | – |
| | INT4 | ✗ | ✗ | **0.962** | 0.345 | 16.3 |
| | | ✗ | ✓ | **0.962** | 0.317 | 16.8 |
| | | ✓ | ✗ | 0.957 | 0.289 | 17.6 |
| FLUX.1-schnell | | ✓ | ✓ | 0.951 | **0.258** | **18.3** |
| | NVFP4 | ✗ | ✗ | 0.957 | 0.280 | 17.5 |
| | | ✗ | ✓ | 0.956 | 0.247 | 18.5 |
| | | ✓ | ✗ | **0.968** | 0.247 | 18.4 |
| | | ✓ | ✓ | **0.968** | **0.227** | **19.0** |
| | BF16 | – | – | 0.944 | – | – |
| | INT4 | ✗ | ✗ | -1.226 | 0.762 | 9.1 |
| | | ✗ | ✓ | -0.902 | 0.763 | 9.9 |
| | | ✓ | ✗ | 0.858 | 0.356 | 17.0 |
| PixArt-Σ | | ✓ | ✓ | **0.878** | **0.323** | **17.6** |
| | NVFP4 | ✗ | ✗ | 0.660 | 0.517 | 14.8 |
| | | ✗ | ✓ | 0.696 | 0.480 | 15.6 |
| | | ✓ | ✗ | 0.945 | 0.290 | 18.0 |
| | | ✓ | ✓ | **0.940** | **0.271** | **18.5** |
| | BF16 | – | – | 0.952 | – | – |
| | INT4 | ✗ | ✗ | 0.894 | 0.339 | 15.3 |
| | | ✗ | ✓ | 0.881 | 0.288 | 16.4 |
| | | ✓ | ✗ | 0.922 | 0.234 | 17.4 |
| SANA-1.6B | | ✓ | ✓ | **0.935** | **0.220** | **17.8** |
| | NVFP4 | ✗ | ✗ | 0.932 | 0.237 | 17.3 |
| | | ✗ | ✓ | 0.927 | 0.202 | 18.3 |
| | | ✓ | ✗ | **0.957** | 0.188 | 18.7 |
| | | ✓ | ✓ | 0.955 | **0.177** | **19.0** |

In Table 3, we provide additional quantitative ablation results of SVDQuant on the MJHQ prompt set (Li et al., 2024a). Across all models, NVFP4 outperforms INT4 due to its native support for smaller microscaling group sizes on Blackwell. SVDQuant leverages a low-rank branch to absorb outliers, significantly enhancing image quality in all settings. Additionally, it can incorporate GPTQ (Frantar et al., 2023) instead of round-to-nearest for weight quantization, further improving quality in most cases. Notably, combining SVDQuant with NVFP4 precision achieves the best results, reaching a PSNR of 21.5 on FLUX.1-dev, closely matching the image quality of the original 16-bit model. In Figure 17, we provide qualitative comparisons across different precision settings.

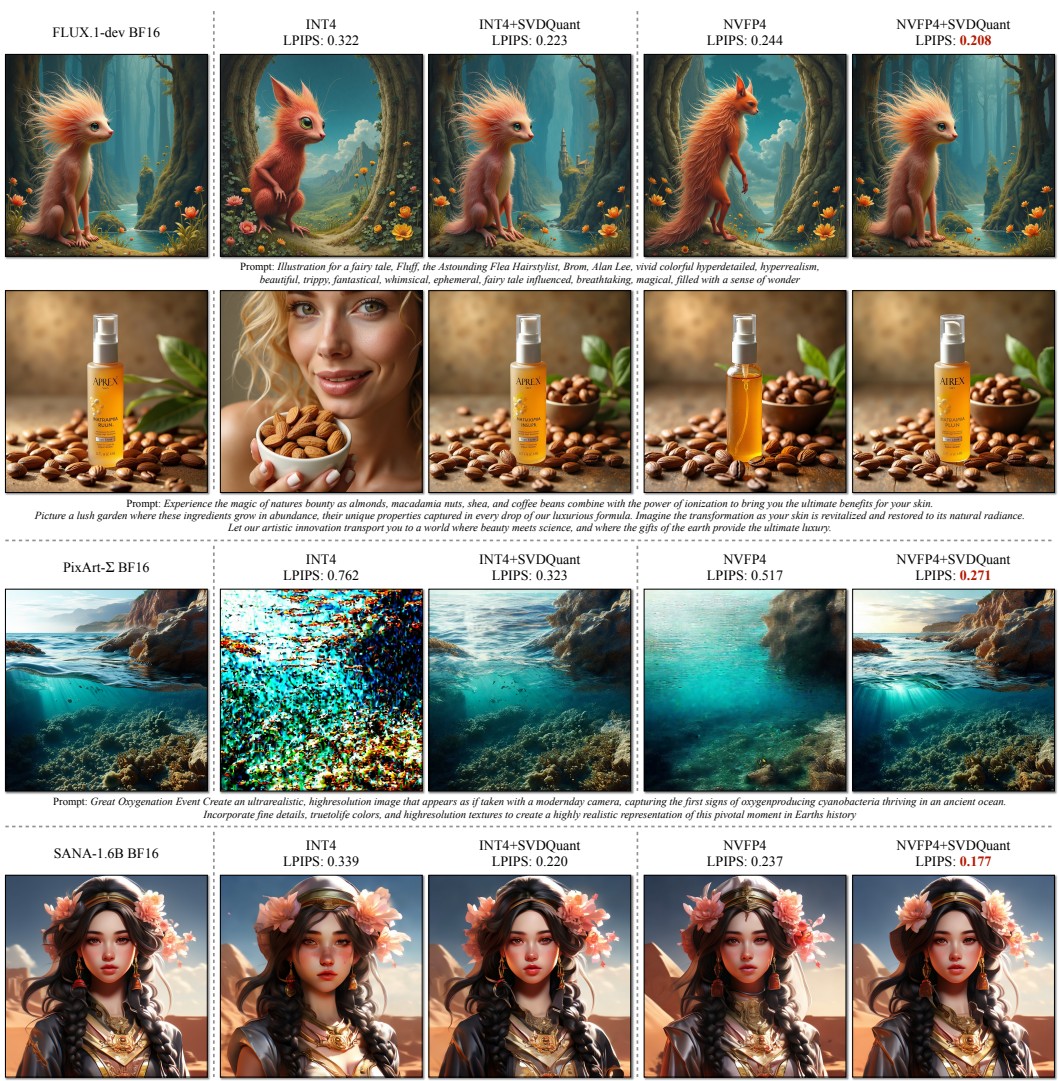

Figure 17: Qualitative comparisons of different precisions on MJHQ. NVFP4+SVDQuant yields the highest image fidelity.

### E.4 LATENCY RESULTS

In Table 4, we compare FLUX latency on a laptop-level 4090 GPU across different precisions. Compared to INT8, 4-bit quantization delivers a $1.3\times$ speedup. However, without optimization, SVDQuant incurs an 18% overhead due to the low-rank branch. By eliminating redundant memory access, Nunchaku achieves latency comparable to naive INT4.

Table 4: Single-step latency comparisons of FLUX on a desktop-level 4090 GPU.

| Method | BF16 | INT8 | Naïve INT4 | SVDQuant | SVDQuant +Nunchaku |
|---|---|---|---|---|---|
| Latency (ms) | 657 | 282 | 212 | 250 | 218 |

### E.5 TRADE-OFF OF INCREASING RANK

Figure 18 presents the results of different rank $r$ in SVDQuant on PixArt-$\Sigma$. Increasing the rank from 16 to 64 significantly enhances image quality but increases parameter and latency overhead. In our experiments, we select a rank of 32, which offers a decent quality with minor overhead.

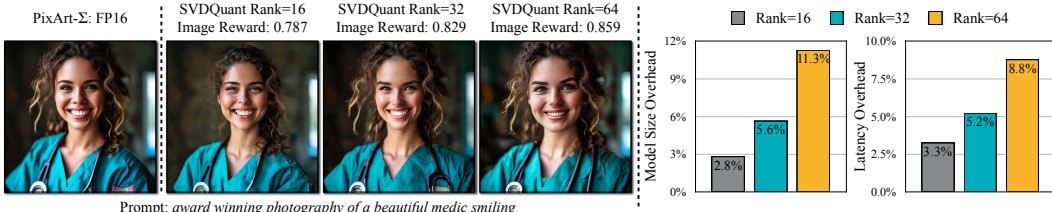

Figure 18: Increasing the rank $r$ of the low-rank branch in SVDQuant can enhance image quality, but it also leads to higher parameter and latency overhead.

### E.6 TRADE-OFF BETWEEN QUALITY AND BITWIDTH

We evaluate LPIPS across different bitwidths for various quantization methods on PixArt-$\Sigma$ and FLUX.1-schnell using the MJHQ dataset in Figure 19, with weights and activations sharing the same bitwidth. Following the convention (Xiao et al., 2023; Lin et al., 2024; 2025; Li et al., 2023a; Zhao et al., 2024d; Dettmers et al., 2022), for bitwidths above 4, we apply per-channel quantization; for 4 or below, we use per-group quantization (group size 64). SVDQuant consistently outperforms naive quantization and SmoothQuant. Notably, on PixArt–$\Sigma$ and FLUX.1-schnell, our 4-bit results match 7-bit and 6-bit naive quantization, respectively.

Our SVDQuant can still generate images in the 3-bit settings on both PixArt-$\Sigma$ and FLUX.1-schnell, performing much better than SmoothQuant. Below this precision (e.g., W2A4 or W4A2), SVDQuant cannot produce images either since 2-bit symmetric quantization is essentially a ternary quantization. Prior work (Ma et al., 2024a; Wang et al., 2023a) has shown that ternary neural networks require quantization-aware training even for weight-only quantization to adapt the weights and activations to the low-bit distribution.

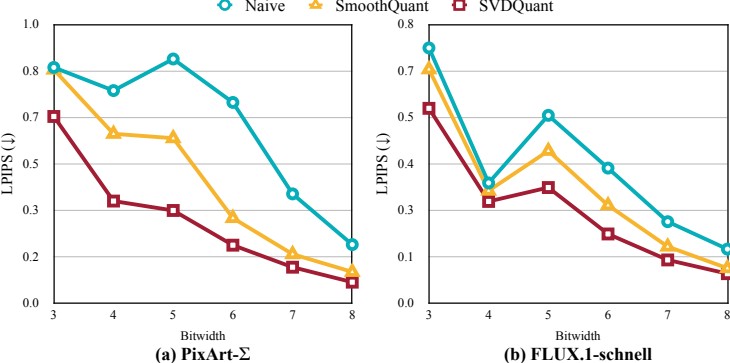

Figure 19: LPIPS of different quantization methods on PixArt-$\Sigma$ and FLUX.1-schnell across different bitwidths.

## F  TEXT PROMPTS

Below we provide the text prompts we use in Figure 9 (from left to right).

```
a man in armor with a beard and a sword
GHIBSKY style, a fisherman casting a line into a peaceful village lake
↪    surrounded by quaint cottages
girl, neck tuft, white hair, sheep horns, blue eyes, nm22 style
sketched style, A squirrel wearing glasses and reading a tiny book under
↪    an oak tree
a panda playing in the snow, yarn art style
```

The text prompts we use in Figure 16 are (in the rasterizing order):

```
A male secret agent in a tuxedo, holding a gun, standing in front of a
↪    burning building
A handsome man in a suit, 25 years old, cool, futuristic
A knight in shining armor, standing in front of a castle under siege
A knight fighting a fire-breathing dragon in front of a medieval castle,
↪    flames and smoke
A male wizard with a long white beard casting a lightning spell in the
↪    middle of a storm
A young woman with long flowing hair, standing on a mountain peak at dawn,
↪    overlooking a misty valley

GHIBSKY style, a cat on a windowsill gazing out at a starry night sky and
↪    distant city lights
GHIBSKY style, a quiet garden at twilight, with blooming flowers and the
↪    soft glow of lanterns lighting up the path
GHIBSKY style, a serene mountain lake with crystal-clear water,
↪    surrounded by towering pine trees and rocky cliffs
GHIBSKY style, an enchanted forest at night, with glowing mushrooms and
↪    fireflies lighting up the underbrush
GHIBSKY style, a peaceful beach town with colorful houses lining the
↪    shore and a calm ocean stretching out into the horizon
GHIBSKY style, a cozy living room with a view of a snow-covered forest,
↪    the fireplace crackling and a blanket draped over a comfy chair

a dog wearing a wizard hat, nm22 anime style
a girl looking at the stars, nm22 anime style
a fish swimming in a pond, nm22 style
a giraffe with a long scarf, nm22 style
a bird sitting on a branch, nm22 minimalist style
a girl wearing a flower crown, nm22 style

sketched style, A garden full of colorful butterflies and blooming
↪    flowers with a gentle breeze blowing
sketched style, A beach scene with kids building sandcastles and seagulls
↪    flying overhead
sketched style, A hot air balloon drifting peacefully over a patchwork of
↪    fields and forests below
sketched style, A sunny meadow with a girl in a flowy dress chasing
↪    butterflies
sketched style, A little boy dressed as a pirate, steering a toy ship on
↪    a small stream
sketched style, A small boat floating on a peaceful lake, surrounded by
↪    trees and mountains

a hot air balloon flying over mountains, yarn art style
a cat chasing a butterfly, yarn art style
a squirrel collecting acorns, yarn art style
a wizard casting a spell, yarn art style
a jellyfish floating in the ocean, yarn art style
a sea turtle swimming through a coral reef, yarn art style
```

