# OpenReview forum: "SVDQuant: Absorbing Outliers by Low-Rank Component for 4-Bit Diffusion Models"
_ICLR.cc/2025/Conference — ICLR 2025 Spotlight_

### Official Review · Reviewer_KPne · 2024-10-24

**Soundness:** 3
**Presentation:** 2
**Contribution:** 2
**Rating:** 6
**Confidence:** 3

**Summary:**

The authors propose SVDQuant to enable 4-bit quantization for diffusion models in terms of both weights and activations, with SVD approach utilized to enable quantization of residual (R=W-L1L2) rather than directly quantizing weight matrices. SVDQuant first use a smoothing technique proposed in previous work to transfer the outliers in activations to weights, then use SVD to enable 16-bit low-rank approximation of the weights and quantize the residual between the two weights, absorbing the weight outliers in the low rank branches L1 and L2.

**Strengths:**

1. I like the idea of translating the weight quantization into residual quantization to eliminate outliers in weights.
2. The figures are well illustrated, and the math presentations are insightful.
3. The model with a dinosaur head is hilarious.

**Weaknesses:**

1. There are a few strong statements in this work with insufficient reasoning.
2. The method section mainly focuses on justifying the minimization of quantization error but lacking discussion of the computation flow.
3. The residual quantization approach looks similar to the quantization of error matrix in LoRC.

**Questions:**

1. Some strong statements need to be properly supported by evidence:

- "Weight-only quantization *cannot* accelerate diffusion models" - For modern GPUs, maybe, but this lacks concrete evidence. Also, different hardware platforms have different bottlenecks.

 - "Weights and activations *must* be quantized to the same bit width" - Can custom hardware support direct mixed precision operations?

 - "they primarily consider weight-only quantization..." For example, the authors have cited Q-diffusion and EfficientDM which quantize both activations and weights. This statement needs further justification.

2. The authors mention that the "lower-precision side will be upcast during computation, negating potential performance boosts". However, as illustrated in Figure 6(b), the XL1L2 branch appears to retain 16-bit full precision before being combined with the quantized residual. This approach seems to be different from the authors' initial claim.

3. It would be very helpful if the authors could elaborate on the similarities and differences between their methods and LoRC (Yao et al.)

Minor:

- How is quantization level defined in Figure 3? And why can it take fractional numbers? I vaguely understand after smoothing, the peak of |X| drops from 10 to 2, but why the peak of |W| also drops? I thought smaller peaks means easier to quantize.

- QKV projection, presumably, is an LLM concept, and it may look abrupt in 4.3 without preliminary discussion.

---

> ### Author Response · Authors · 2024-11-23
> **Author Response to Reviewer KPne**
>
> ### Strong Statements
> > "Weight-only quantization cannot accelerate diffusion models" - For modern GPUs, maybe, but this lacks concrete evidence. Also, different hardware platforms have different bottlenecks.
>
> We are referring to modern GPUs. As shown in Figure 8 of our original submission, NF4 weight-only quantization achieves the same latency as the original BF16 model on desktop-level 4090 GPUs. Similarly, Figure 3 in QServe [1] shows that W4A16 (green lines) and W16A16 (red lines) converge to the same peak performance at large batch sizes. Diffusion models process thousands of tokens in parallel, effectively resulting in a large batch size. We have updated our manuscript to provide this GPU context.
>
> > "Weights and activations must be quantized to the same bit width" - Can custom hardware support direct mixed precision operations?
>
> Yes, specialized hardware may support mixed-precision operations. We have revised our manuscript to limit the scope to modern GPUs. Thanks for your feedback.
>
> > "they primarily consider weight-only quantization..." For example, the authors have cited Q-diffusion and EfficientDM which quantize both activations and weights. This statement needs further justification.
>
> Thanks for your feedback. We refer to methods like QLoRA [2] here, which fine-tune LoRA branches on a quantized base model. Q-Diffusion was already cited earlier in the previous paragraph, so it was not included here. We've revised our manuscript to clarify this claim.
>
> ### Computation Flow
> In our original submission, we have provided an overview of our method in Figure 3 and illustrated the computation flow in Figure 6(b). Could you advise on any additional computation flows that should be included in the manuscript?
>
> ### Comparison with LoRC
> In our original submission, we discussed LoRC in Section 2 (Low-rank decomposition) and compared it in Section 5.2 (Ablation study) and Figure 9. We highlight the results below. Our SVDQuant significantly outperformed LoRC. Specifically, LoRC uses a low-rank branch to compensate quantization errors, but this approach is suboptimal. Quantization errors resemble white noise with a smooth singular value distribution, making low-rank compensation less effective.
> In contrast, SVDQuant decomposes weights and quantizes only the residual. As shown in Figure 5, the first few singular values dominate, enabling us to shift them to the low-rank branch. This reduces weight magnitude and eases the weight quantization. Furthermore, by migrating activation outliers into the weights, SVDQuant lowers activation quantization difficulty, resulting in better image quality.
>
> | Method | FID (↓) | Image Rewad (↑) | LPIPS (↓) | PSNR (↑) |
> |---|---|---|---|---|
> | LoRC | 180 | -0.965 | 0.754 | 9.67 |
> | **SVDQuant** | **20.1** | **0.898** | **0.394** | **16.2** |
>
> ### Clarification on the Upcast
> This is a misunderstanding. The sentence refers to quantization without the low-rank branch, where "side" refers to either weights or activations. For example, in W4A16 weight-only quantization on GPUs, weights are upcast to 16 bits during computation, meaning the operations still use 16-bit multiplication. We have revised our manuscript to make this clearer.
>
> ### Clarification on Figure 3
> Quantization level refers to the number of distinct values that can be represented. For example, 3-bit quantization allows for 8 distinct values, so our figures show 8 level lines. The top-left numbers indicate the largest representable values, which can be fractional numbers. After smoothing, the peak value of $|{\hat{\boldsymbol{W}}}|$ should be 4.5, not 0.5—this was a typo, and we have corrected it in our revision. Yes, smaller peaks mean easier to quantize.
>
> ### QKV Projections
> Thank you for your suggestion. Recent diffusion models include transformer blocks to enhance performance. The QKV projections refer to the layers within these transformer blocks. We have updated our manuscript to clarify this concept.
>
> #### References
> 1. QServe: W4A8KV4 Quantization and System Co-design for Efficient LLM Serving
> 2. QLoRA: Efficient Finetuning of Quantized LLMs

---

> > ### Author Response · Authors · 2024-11-28
> > **Follow-Up on Author Response**
> >
> > Dear Reviewer KPne,
> >
> > It has been five days since we submitted our Author Response addressing your concerns. We understand your tight schedule, but given that the discussion phase is approaching its end, we sincerely hope that you can update your assessment if our response addresses your questions. If you have any other questions, we are willing to discuss that with you as well.
> >
> > Thank you once again for your time and effort in reviewing our paper. We wish you a wonderful Thanksgiving!
> >
> > Best regards, The Authors

---

> > > ### Author Response · Authors · 2024-12-02
> > > **Final Day of the Discussion Phase**
> > >
> > > Dear Reviewer KPne,
> > >
> > > Hope you had a wonderful thanksgiving!
> > >
> > > As we approach the final day of the discussion phase, we wonder if you could kindly consider updating the score if our responses have addressed your concerns. If you have any remaining questions or need further clarification, we would be happy to discuss them with you.
> > >
> > > Thank you once again for your valuable valuable efforts in reviewing our paper.
> > >
> > > Best regards, The Authors

---

### Official Review · Reviewer_44PJ · 2024-11-02

**Soundness:** 3
**Presentation:** 4
**Contribution:** 3
**Rating:** 8
**Confidence:** 4

**Summary:**

The paper proposes an aggressive quantization method (4-bit) for both activations and weight in diffusion models. To mitigate quantization-induced quality losses, the author propose to  migrate outlayers from activations to weights and then to add low-rank branches at high accuracy 16b (as opposed to the traditional smoothing approach which is not sufficient at 4-bit quantization level especially for diffusion models): i.e. the network is modified also for inference by the presence of the branches A baseline implementation of the low-rank branches in inference is slow and negates the speed advantages of the quantized computations, The authors then propose a remedy to this shortcoming which merges the branches, to regain speed (while also retaining memory reduction advantages)

**Strengths:**

The paper presents clearly the flow of ideas: statements are clear and detailed proofs are moved to the appendix, hence they do not distract from the flow f the key messages.
The motivation for the work is clearly described as well as the gaps in the SoA, and the claims of novelty.
The reader is led step by step to the final solution: this flow helps understanding the technical motivations of the various steps.
Experiments are clear and the KPI used are well defined.

**Weaknesses:**

It is not clear ho lambda for the migration of outlayers from activations and weights si computed (it is per-channel - so the array can be quite big). This is in my view a bit of a problem, because lamda migh be impacted by the dataset used for calibration (if it is decided offline) or may be hard to determine efficiently online.

Inference time results are missing because the authors have no access to modern gpus (Blackwell) with native 4b support. This is a minor weakness, but it must be noted that it's not fully clear if the merged kernels will benefit from the same speedup claimed for pure 4b kernels.

the key ideas are not novel per se, but the combination is interesting - the key intuition being the merged low-rank and low-precision kernel to recover speed

**Questions:**

Please clarify how the diag lambda is determined
if possible provide an estimate of the speedup expected by the mixed kernel - possibly using mixed kernel ad higher precision and see how the speedup degrades from "pure" low precision kernel.

---

> ### Author Response · Authors · 2024-11-23
> **Author Response to Reviewer 44PJ**
>
> Thanks for supporting our paper. Below, we address each of your questions individually.
>
> ### Lambda Selection
>
> The smoothing factor $\lambda \in \mathbb R^{m}$ is a per-channel vector whose $i$-th element can be computed as $\lambda_i = \max(|{\boldsymbol{X}} _{:,i}|)^\alpha / \max(|{\boldsymbol{W}} _{i,:}|)^{1-\alpha}$ following SmoothQuant [1]. Here, ${\boldsymbol{X}} \in \mathbb R^{b\times m}$ and ${\boldsymbol{W}} \in \mathbb R^{m\times n}$. It is decided offline by searching for the best migration strength $\alpha$ for each layer to minimize the layer output mean squared error (MSE) after SVD on the calibration dataset. We have included this in Appendix B.
>
> ### Blackwell Performance
> Blackwell GPUs were not on the market at the time of submission. However, our fused kernel is expected to achieve similar or better speedup on these GPUs. The advantage of kernel fusion lies in reducing memory access, making it independent of the specific computation instructions. As long as the ratio of peak performance to peak memory bandwidth remains unchanged, we anticipate similarly minimal overhead. Furthermore, with native microscaling group quantization support, we can anticipate greater speedups compared to our INT4 results on 4090 GPUs.
>
> ### Novelty
> Thanks for your interest in our method.
>
> ### Speedup Compared to Pure Low-Bit Kernels
> On the laptop-level 4090 GPU, naive INT4 FLUX requires 212ms for a single forward. SVDQuant introduces low-rank branches, increasing latency to 250ms without optimization (18% overhead). LoRunner effectively cuts off redundant memory access and achieves 218ms latency, adding just 3% overhead.

---

> > ### Author Response · Authors · 2024-12-02
> > **Final Day of the Discussion Phase**
> >
> > Dear Reviewer 44PJ,
> >
> > We hope you had a wonderful thanksgiving!
> >
> > As we approach the final day of the discussion phase, we wonder if you could kindly consider updating the score if our responses have addressed your concerns. If you have any remaining questions or need further clarification, we would be happy to discuss them with you.
> >
> > Thank you once again for supporting our paper and for your invaluable efforts in reviewing it.
> >
> > Best regards, The Authors

---

### Official Review · Reviewer_2VPm · 2024-11-04

**Soundness:** 2
**Presentation:** 3
**Contribution:** 3
**Rating:** 6
**Confidence:** 4

**Summary:**

This work accelerate diffusion models by low-bit quantization and low-rank decomposition. The motivation of this work is the outlier quantization error in activations for recent PTQ methods. The authors propose a multi-branch architecture with both low-bit and low-rank operations. To speed up the inference, a kernel fusion implementation is also proposed for the two branches. In experiments, the DiT and UNet diffusion models are evaluated in 4/8-bit conditions.

**Strengths:**

- It is an interesting topic to combine the low-bit quantization and low-rank decomposition, previous work including Loft-Q (ICLR 24), et al.

- This work proposes a kernel fusion implementation to speed up on-device inference.

- The method is reasoning and the writing is easy to follow.

**Weaknesses:**

- I don't think splitting low-rank and low-bit branches is a good idea to overcome quantization errors. Different from the QLLM (ICLR 24) et al.,  the computation of the two branches can not merge after training, leading to 5~10% overheads in the paper (line 314).

- Experiments parts are not very solid: the baseline quantization methods are kind-of weak. Recent works including SpinQuant [1], AffineQuant [2], et al. achieve much higher performance than the baseline NF4 in the paper and also without the additional branch.

[1] SpinQuant: LLM quantization with learned rotations
[2] AffineQuant: Affine Transformation Quantization for Large Language Models

**Questions:**

In the experiment part, this work almost evaluates on 4/8-bit quantization; what is the best trade-off between bit-width and performance for diffusion models?

Also, adding more best practice of the diffusion model quantization will be fine.

---

> ### Comment · Reviewer_2VPm · 2024-11-23
>
> Thanks for the general reply, and I have increased my score. Moreover, what is the setting of QuaRot in your experiments, RTN or GPTQ for weight quantization?

---

> > ### Author Response · Authors · 2024-11-23
> > **Follow-Up on Reviewer Reply**
> >
> > Thank you for your prompt response and for raising the score. I have uploaded our detailed reply to your review. Could you kindly take a look and let us know if it addresses your concerns? Please don't hesitate to reach out if there are any additional clarifications or experiments we can provide.
> >
> > Regarding your question about the QuaRot setting, I need to confirm with my collaborator and will follow up with you shortly.

---

> > ### Author Response · Authors · 2024-11-25
> > **Follow-up on the QuaRot setting**
> >
> > Hi Reviewer 2VPm,
> >
> >   In our previous experiments, we applied RTN to both QuaRot and SVDQuant for fair comparisons. We further apply the GPTQ to both methods and show the results on PixArt-∑ below. Specifically, SVDQuant+RTN still outperforms QuaRot with either RTN or GPTQ consistently. Remarkably, as discussed in the previous response, **QuaRot also incurs a significant 21% measured latency overhead** due to the online rotation. When GPTQ is applied, SVDQuant achieves even better FID, LPIPS, and PSNR.
> >
> >
> >
> > |   Method             |   FID (↓)   |   Image Rewad (↑)  |   LPIPS (↓)  |   PSNR (↑)  |
> > |----------------------|-------------|--------------------|--------------|-------------|
> > |   QuaRot+RTN         |   28.2      |   0.847            |   0.459      |   15.3      |
> > |   QuaRot+GPTQ   |   26.6      |   0.877            |   0.433      |   16.2      |
> > |   **SVDQuant+RTN**       |   20.1      | **0.898**       |   0.394      |   16.2      |
> > |   **SVDQuant+GPTQ**      | **19.4** |   0.877            | **0.327** | **17.5** |
> >
> >   As the discussion phase is coming to an end soon, we warmly welcome any additional comments or suggestions you may have. If our responses have addressed your concerns, could you please consider upgrading the score. Thank you very much!
> >
> > Best regards, The Authors

---

> > > ### Author Response · Authors · 2024-12-02
> > > **Final Day of the Discussion Phase**
> > >
> > > Dear Reviewer 2VPm,
> > >
> > > Hope you had a wonderful thanksgiving!
> > >
> > > As we approach the final day of the discussion phase, we wonder if you could kindly consider updating the score if our responses to your review and the above clarification on the QuaRot setting have addressed your concerns. If you have any remaining questions or need further clarification, we would be happy to discuss them with you.
> > >
> > > Thank you once again for your valuable valuable efforts in reviewing our paper.
> > >
> > > Best regards, The Authors

---

> ### Author Response · Authors · 2024-11-23
> **Author Response to Reviewer 2VPm**
>
> Thanks for raising the score. Below, we address each of your questions individually.
> ### Bitwidth-Performance Tradeoff
> We evaluate LPIPS across different bitwidths for various quantization methods on PixArt-∑ and FLUX.1-schnell using the MJHQ dataset, with weights and activations sharing the same bitwidth. Following the convention [1-6], for bitwidths above 4, we apply per-channel quantization; for 4 or below, we use per-group quantization (group size 64). SVDQuant consistently outperforms naive quantization and SmoothQuant. Notably, on PixArt-∑ and FLUX.1-schnell, our 4-bit results match 7-bit and 6-bit naive quantization, respectively. See Figure 18 in our revised manuscript for the tradeoff curve.
> | Model | Bitwidth | 8 | 7 | 6 | 5 | 4 | 3 |
> |---|---|---|---|---|---|---|---|
> | PixArt-∑ | Naive | 0.210 | 0.392 | 0.720 | 0.876 | 0.762 | 0.845 |
> |  | SmoothQuant | 0.112 | 0.176 | 0.305 | 0.592 | 0.607 | 0.838 |
> |  | **SVDQuant (Ours)** | **0.075** | **0.129** | **0.208** | **0.332** | **0.394** | **0.669** |
> | FLUX.1-schnell | Naive | 0.155 | 0.234 | 0.387 | 0.539 | 0.345 | 0.845 |
> |  | SmoothQuant | 0.100 | 0.162 | 0.281 | 0.437 | 0.323 | 0.838 |
> |  | **SVDQuant (Ours)** | **0.084** | **0.124** | **0.198** | **0.332** | **0.292** | **0.669** |
>
> ### Low-Rank Branch Overhead
> Introducing a low-rank branch significantly improves the tradeoff between image quality and bitwidth in SVDQuant. For example, achieving the same quality as SVDQuant’s W4A4 requires SmoothQuant to use approximately 6 bits as shown in the above tradeoff, resulting in a ~50% overhead. In contrast, with LoRunner, the low-rank branch adds only a 3% overhead for a single DiT forward pass (see Table 4 in our revised manuscript). Additionally, SVDQuant consistently outperforms QLLM on both PixArt-∑ and FLUX.1-schnell, with a particularly notable advantage on PixArt-∑, as shown in the below table.
>
> Moreover, users often apply custom low-rank adapters (LoRAs) when using diffusion models. LoRunner eliminates redundant memory access for the low-rank branch, enabling it to be retained during inference. This allows seamless support to LoRAs by appending their weights to the low-rank branch, which avoids the need for weight fusion and re-quantization. Figure 9 and 17 in our revised manuscript show some visual results of applying five different LoRAs to our INT4 FLUX.1-dev, matching the image quality of the original 16-bit model.
>
> | Model | Method | FID (↓) | Image Rewad (↑) | LPIPS (↓) | PSNR (↑) |
> |---|---|---|---|---|---|
> | PixArt-∑ | Naive | 206 | -1.23 | 0.762 | 9.08 |
> |  | SmoothQuant | 48.6 | 0.617 | 0.607 | 12.9 |
> |  | QLLM | 35.8 | 0.763 | 0.581 | 13.1 |
> |  | AffineQuant | 29.6 | 0.816 | 0.540 | 14.5 |
> |  | QuaRot | 28.2 | 0.847 | 0.459 | 15.3 |
> |  | **SVDQuant (Ours)** | **20.1** | **0.898** | **0.394** | **16.2** |
> | FLUX.1-schnell | Naive | 18.1 | 0.962 | 0.345 | 16.3 |
> |  | SmoothQuant | 18.4 | 0.943 | 0.323 | 16.7 |
> |  | QLLM | 18.3 | 0.959 | 0.295 | 17.3 |
> |  | AffineQuant | 22.8 | 0.937 | 0.292 | 16.9 |
> |  | QuaRot | 19.3 | 0.951 | 0.287 | 17.4 |
> |  | NF4 (W4A16) | 18.9 | 0.943 | **0.257** | **18.2** |
> |  | **SVDQuant (Ours)** | **18.1** | **0.965** | 0.292 | 17.5 |
>
> ### Comparisons with LLM Baselines
> NF4 is a strong baseline in terms of image quality, which uses W4A16 precision instead of W4A4. At the reviewer’s request, we adapted LLM quantization methods (QLLM, QuaRot, and AffineQuant) to diffusion models and show W4A4 results above on MJHQ. **SpinQuant is not applicable due to the high overhead of rotation-based methods, as explained in Section 3 of our original submission.** These methods require fusing layer norm weights into the following linear layer and fusing rotation matrix into both the previous and following linear layers. However, Diffusion Transformers (DiTs) use adaptive layer norms whose weights are dynamically generated based on the input condition. This prevents the rotation matrix from being merged into the previous linear layer. Therefore, rotation-based methods must perform online rotation, which adds an extra full-rank matrix multiplication for SpinQuant and results in a **prohibitive 730% measured latency overhead** on FLUX.1 models. As an alternative, we evaluate another well-known rotation-based method QuaRot, which exploits Hadamard transformation as rotation. Even optimized with the SoTA  `fast_hadamard_transform` package, **QuaRot incurs a 21% overhead**, much higher than ours.
>
> SVDQuant outperforms all baselines by a wide margin across all metrics on PixArt-∑. On FLUX.1-schnell, SVDQuant achieves the best FID and Image Reward scores and ranks second in PSNR only behind NF4.

---

> > ### Author Response · Authors · 2024-11-23
> > **Author Response to Reviewer 2VPm (Continued)**
> >
> > #### References
> > 1. SmoothQuant: Accurate and Efficient Post-Training Quantization for Large Language Models
> > 2. LLM.int8(): 8-bit Matrix Multiplication for Transformers at Scale
> > 3. Q-Diffusion: Quantizing Diffusion Models
> > 4. Atom: Low-bit Quantization for Efficient and Accurate LLM Serving
> > 5. AWQ: Activation-aware Weight Quantization for LLM Compression and Acceleration
> > 6. GPTQ: Accurate Post-Training Quantization for Generative Pre-trained Transformers

---

### Official Review · Reviewer_ntPk · 2024-11-05

**Soundness:** 4
**Presentation:** 4
**Contribution:** 3
**Rating:** 8
**Confidence:** 4

**Summary:**

The paper introduces a post-training quantization method for diffusion models called SVDQuant that successfully quantizes both weights and activations to 4 bits without sacrificing visual quality. By integrating the low-rank branch kernels into the low-bit branch, SVDQuant minimizes overhead, significantly accelerating model inference. Tested on the 12B parameter FLUX.1-schnell model, it reduces memory usage by 3.6 times compared to the BF16 model and achieves a 3.6 times speedup over the NF4 W4A16 baseline on a laptop equipped with an RTX-4090 GPU.

**Strengths:**

1. The paper introduces a full-precision low-rank adapter and Singular Value Decomposition (SVD) to effectively compensate for quantization errors.
2. The authors have implemented a novel kernel that efficiently fuses low-rank and low-bit branches, minimizing computational overhead.
3. The extensive experiments conducted on state-of-the-art diffusion models, such as FLUX, provide strong evidence of the method's effectiveness and robustness.

**Weaknesses:**

1. Adding a W8A8 baseline to the latency comparison would provide valuable insights and a clearer performance reference.
2. Since the authors use group quantization for weights and activations in the 4-bit setting, it would be beneficial to include methods that also use group quantization, such as Atom [1], as baselines.

References:
[1] Zhao et al. "Atom: Low-bit Quantization for Efficient and Accurate LLM Serving" in MLSys'24

**Questions:**

1. Could you explain the calculation method for the smoothing factor in more detail?
2. In the ablation study, what is meant by "SVD-only" and "naive quantization"? Does "SVD-only" indicate that no quantization is applied? And what is the setting for naive quantization?

---

> ### Author Response · Authors · 2024-11-23
> **Author Response to Reviewer ntPk**
>
> Thanks for supporting our paper. Below, we address each of your questions individually.
> ### 8-Bit Latency Comparison
> Below we compare the FLUX latency of the 8-bit and our 4-bit models on a desktop-level 4090 GPU. Our 4-bit model achieves approximately 1.3× speedup over the 8-bit model. We have correspondingly updated the results in our revision (see Appendix D.3 and Table 4).
> | Precision | Latency (ms) |
> |---|---|
> | INT8 | 282 |
> | SVDQuant INT4 | 218 |
>
> ### Comparison with Atom
> As suggested by the reviewer, we compared SVDQuant with Atom on FLUX.1-schnell using the MJHQ dataset. In the INT W4A4 setting, SVDQuant outperforms Atom across all metrics. Additionally, on a desktop-level 4090 GPU, SVDQuant with LoRunner is 1.2× faster than Atom.
> | Method | FID (↓) | Image Reward (↑) | LPIPS (↓) | PSNR (↑) | Latency (ms) |
> |---|---|---|---|---|---|
> | Atom | 18.4 | 0.943 | 0.323 | 16.7 | 261 |
> | **SVDQuant (Ours)** | **18.1** | **0.965** | **0.292** | **17.5** | **218** |
>
> ### Smoothing Process
> The smoothing factor $\lambda \in \mathbb R^{m}$ is a per-channel vector whose $i$-th element can be computed as $\lambda_i = \max(|{\boldsymbol{X_{:,i}}}|)^\alpha / \max(|{\boldsymbol{W_{i,:}}}|)^{1-\alpha}$ following SmoothQuant [1]. Here, ${\boldsymbol{X}} \in \mathbb R^{b\times m}$ and ${\boldsymbol{W}} \in \mathbb R^{m\times n}$. It is decided offline by searching for the best migration strength $\alpha$ for each layer to minimize the layer output mean squared error (MSE) after SVD on the calibration dataset. We have included this in Appendix B.
>
> ### Ablation Setting
> Yes, "SVD-only" refers to applying SVD to the weights and keeping only the low-rank component without quantization. "Naive quantization" refers to directly applying per-group quantization without any smoothing.
>
> #### References
> 1. SmoothQuant: Accurate and Efficient Post-Training Quantization for Large Language Models

---

> > ### Comment · Reviewer_ntPk · 2024-11-26
> >
> > Thank you for your thoughtful rebuttal. I maintain my rating.

---

### Official Review · Reviewer_2HuL · 2024-11-06

**Soundness:** 3
**Presentation:** 3
**Contribution:** 4
**Rating:** 8
**Confidence:** 3

**Summary:**

This paper proposes a 4-bit PTQ method for diffusion models that absorbs the outliers between weights and activations using a low-rank branch. To augment this method, the paper also presents an efficient inference engine to avoid the redundant memory access of the activations. Experimental results on latest diffusion models validate the superior accuracy and memory efficiency of the proposed method.

**Strengths:**

The paper is well-written and clear. Though the idea is generally intuitive, the observation of the low-rank outliers and the fusion from the low-rank branch to low-bit branch makes this work a strong submission. Experiments are conducted on the latest diffusion models, and the results are generally strong.

**Weaknesses:**

1. Can the authors include some results on sub 4-bit quantization with the proposed method? It would be good to know the limitations of the method.

2. The authors should include more comparisons in their experimental results. While they compared with MixDQ and ViDiT-Q, some prominent works have been left out, such as Q-Diffusion [1] and QUEST [2].

[1] https://arxiv.org/pdf/2302.04304

[2] https://arxiv.org/pdf/2402.03666v1

**Questions:**

Please see above.

---

> ### Author Response · Authors · 2024-11-23
> **Author Response to Reviewer 2HuL**
>
> Thanks for supporting our paper. Below, we address each of your questions individually.
> ### Results Below 4-Bit Quantization
> Below we show the results of sub-4-bit quantization. Our SVDQuant can still generate images in the 3-bit settings on both PixArt-∑ and FLUX.1-schnell, performing much better than SmoothQuant. Below this precision (e.g., W2A4 or W4A2), SVDQuant cannot produce images either, since 2-bit symmetric quantization is essentially a ternary quantization. Prior work [1,2] has shown that ternary neural networks require quantization-aware training even for weight-only quantization to adapt the weights and activations to the low-bit distribution. We have included the discussion in Appendix D.5 in our revised paper.
> | Model | Precision | FID (↓) | Image Reward (↑) | LPIPS (↓) | PSNR (↑) |
> |---|---|---|---|---|---|
> | PixArt-∑ | Our W4A4 | 20.1 | 0.90 | 0.394 | 16.2 |
> |  | SmoothQuant W3A4 | 401 | -2.25 | 0.784 | 7.83 |
> |  | Our W3A4 | 62.2 | 0.10 | 0.572 | 13.7 |
> |  | Our W3A3 | 176 | -1.38 | 0.669 | 12.5 |
> | FLUX.1-schnell | Our W4A4 | 18.1 | 0.965 | 0.292 | 17.5 |
> |  | SmoothQuant W3A4 | 22.2 | 0.857 | 0.529 | 13.8 |
> |  | Our W3A4 | 19.2 | 0.882 | 0.446 | 15.0 |
> |  | SmoothQuant W3A3 | 42.5 | 0.283 | 0.671 | 13.2 |
> |  | Our W3A3 | 30.4 | 0.677 | 0.559 | 14.0 |
>
> ### Comparisons with Q-Diffusion and QuEST
> In our submission, we demonstrated that SVDQuant outperforms MixDQ and ViDiT-Q (see Table 1 and Figure 7). Both MixDQ and ViDiT-Q have already compared against Q-Diffusion [3] as a baseline and outperformed it in their respective papers (refer to Figure 1, Tables 1 and 2 in MixDQ, and Table 3 in ViDiT-Q). QuEST [4] is a quantization-aware training method that requires additional fine-tuning and was therefore not included in our submission.
>
> At the reviewer's request, we conducted additional experiments comparing these baselines on Stable Diffusion 1.4 using the MJHQ dataset. SVDQuant still achieves significantly better LPIPS and PSNR. We have included the results in Appendix D.3 and Table 4 in our revised manuscript.
> | Method | FID (↓) | LPIPS (↓) | PSNR (↑) |
> |---|---|---|---|
> | Q-Diffusion | 368 | 0.862 | 8.00 |
> | QuEST | **25.1** | 0.771 | 8.48 |
> | **SVDQuant (Ours)** | 38.3 | **0.393** | **15.9** |
>
> #### References
> 1. BitNet: Scaling 1-bit Transformers for Large Language Models
> 2. The Era of 1-bit LLMs: All Large Language Models are in 1.58 Bits
> 3. Q-Diffusion: Quantizing Diffusion Models
> 4. QuEST: Low-bit Diffusion Model Quantization via Efficient Selective Finetuning

---

> > ### Comment · Reviewer_2HuL · 2024-12-01
> >
> > Thanks for the new experimental results, and I am satisfied with your response. I would keep my score, but would have liked to give a score of 9.

---

### Official Review · Reviewer_R3Nw · 2024-11-07

**Soundness:** 3
**Presentation:** 3
**Contribution:** 2
**Rating:** 8
**Confidence:** 3

**Summary:**

This paper presents SVDQuant quantizes weights and activations to 4 bits to accelerate inference of large-scale diffusion models. Through a low-rank decomposition branch, the authors introduce SVDQuant, a method for mitigating the impact of outliers on quantization.
They absorb outliers on weights/activations. To do so, they first migrate the outliers from activation to weight. Then they apply SVD to the updated weight, decomposing it into a low-rank branch and a residual.  Additionally, they incorporate an inference engine, LoRunner, which combines low-rank and low-bit kernels to maximize computation speed and minimize memory overhead. As demonstrated in empirical results for different large diffusion models, SVDQuant offers significant memory savings and performance improvements without affecting image quality significantly. This demonstrates that it is a promising approach for deploying diffusion models on consumer-grade hardware.

**Strengths:**

1-SVDQuant’s strategy to manage outliers using low-rank decomposition offers an improvement over standard smoothing techniques.

2-LoRunner effectively reduces memory access overhead and enhances performance, particularly for GPU inference, addressing practical deployment constraints.

3-Pushing boundaries for low-bit quantization of both weight and activation

**Weaknesses:**

1-The level of contribution is not great. Using outliers and low rank for quantizations is a very well-known technique. The authors need to have a section in related work and highlight the difference/novelty between the technique and all related work. Otherwise, the novelty seems incremental.

2- The paper lacks a theoretical analysis of why the low-rank decomposition approach can consistently outperform other outlier-handling techniques beyond empirical observations.

3- Code is not provided.

**Questions:**

1- Using outliers and low rank for quantizations is a very well-known technique. The authors need to have a section in related work and highlight the difference/novelty between the technique and all related work. Otherwise, the novelty seems incremental.
Maybe you can provide a table for all related work including the mentioned one here and show all similarities and differences with the proposed technique.

[A] QNCD: Quantization Noise Correction for Diffusion Models

[B] Q-DiT: Accurate Post-Training Quantization for Diffusion Transformers


2- The choice of rank (e.g., rank 32) appears somewhat arbitrary, without adequate theoretical or empirical justification for why this setting was chosen over others.  Can the author put more comments about it? Maybe more ablation studies and more detailed experiments about it will give more insight to the reviewer.

3- If a neural network exhibits skewed distributions that may not align well with the low-rank assumption used here (e.g., nongaussian behavior), how does it work? Please discuss the robustness of the method to different weight distributions.

4-With the advent of mixed-precision computation, some architectures might benefit from a mix of 4-, 8-, and. How would LoRunner perform in such configurations, and could it be adapted to handle this flexibility?

5-What are the challenges to extending the method below 4-bit? For example 2w2a or 4w2a or 2w4a?

6-I am interested to see a breakdown of LoRunner’s impact on speed and memory, perhaps by comparing SVDQuant with and without LoRunner.

7- Minor typos:
"Quantizes both the weights and activations" – quantizes both weights and activations.
“on NVIDIA RTX-4090 laptop" – "on an NVIDIA RTX-4090 laptop."

---

> ### Author Response · Authors · 2024-11-23
> **Author Response to Reviewer R3NW**
>
> ### Novelty & Related Work Comparisons
> Previous works combining low-rank decomposition with quantization primarily focus on parameter-efficient fine-tuning (PeFT) such as QLoRA [1], where low-rank branches are fine-tuned on a quantized base model. These methods do not aim to improve quantization performance and only quantize weights. The exception, LoRC [2], uses a low-rank branch to compensate for quantization errors. However, this approach is suboptimal, as quantization errors resemble white noise, which low-rank branches struggle to mitigate effectively, as shown in Section 5.2 (Ablation study) and Figure 9 of our original submission. In contrast, we carefully analyze the sources of quantization errors and propose a principled post-training quantization (PTQ) method to absorb outliers in both weights and activations, significantly reducing errors. Additionally, our inference engine is organic with our quantization approach. Without it, the overhead from low-rank branches would negate the potential speedup, making our method impractical for real-world use on modern GPUs.
>
> As requested by the reviewer, we discuss the similarities and differences between our method and related works, including QNCD [3] and Q-DiT [4] in Table 2 of our revised manuscript. Our method differs distinctly from these works. To avoid being too verbose, we only sample 4 representative works below.
> | Method | Similarity | Difference |
> |---|---|---|
> | QNCD [3] | Both are PTQ methods for diffusion models. | The methods differ entirely. QNCD is only applied to U-Net backbones, while our approach supports both U-Net and DiT. Additionally, we push the boundary quantization from W4A8 to W4A4 and demonstrate speedups on GPUs. |
> | Q-DiT [4] | Both are PTQ methods for diffusion models. | The methods differ entirely. Besides, Q-DiT is only applied to class-conditioned models, while we can work on large text-to-image models. We also push the quantization boundary from their W4A8 to W4A4 and demonstrate speedups on GPUs. |
> | MixDQ [5] | Both are PTQ methods for diffusion models. | The methods differ entirely. Besides, MixDQ is only applied U-Net models, while we can work on both the U-Net and DiT backbones. We also push the quantization boundary from W4A8 to W4A4. |
> | EfficientDM [6] | Both are diffusion quantization methods with low-rank branches. | They use low-rank branches to reduce the cost of quantization-aware training, requiring fusion after tuning. In contrast, our method doesn't need training and preserves the low-rank branches during inference. Additionally, EfficientDP is only applied to class-conditioned U-Nets. We support large text-to-image models and demonstrate speedups on GPUs. |
>
> ### Theoretical Analysis of Why SVDQuant Works Better
> We acknowledge the desire for theoretical analysis. While a comprehensive theory for quantization methods is challenging due to dependencies on data and weight distributions, we did provide meaningful theoretical insights in our paper:
> * Our Proposition 4.1 decomposes quantization error into activation and weight components, which we address through smoothing and low-rank decomposition respectively.
> * Proposition 4.2 shows that quantization error scales with matrix norm. Our low-rank branch reduces this norm, minimizing error and providing a theoretical basis for our strong empirical performance.
>
> Compared to other outlier-handling methods like smoothing and rotation:
> * Smoothing shifts outliers between weights and activations, making activation quantization easier but weight quantization harder, resulting in limited overall error reduction.
> * Rotation redistributes outliers across channels but fails to reduce the matrix norm, leading to higher errors than our approach.
>
> We also provide experimental comparisons with SmoothQuant [7] (smoothing-based) and QuaRot [8] (rotation-based) on the MJHQ dataset under W4A4 settings. As shown in the table below, SVDQuant outperforms both SmoothQuant and QuaRot on PixArt-∑ and FLUX.1-schnell, particularly with a large margin on PixArt-∑.
>
> | Model | Method | FID (↓) | Image Rewad (↑) | LPIPS (↓) | PSNR (↑) |
> |---|---|---|---|---|---|
> | PixArt-∑ | SmoothQuant | 48.6 | 0.617 | 0.607 | 12.9 |
> |  | QuaRot | 28.2 | 0.847 | 0.459 | 15.3 |
> |  | **SVDQuant (Ours)** | **20.1** | **0.898** | **0.394** | **16.2** |
> | FLUX.1-schnell | SmoothQuant | 18.4 | 0.943 | 0.323 | 16.7 |
> |  | QuaRot | 19.3 | 0.951 | 0.287 | 17.4 |
> |  | **SVDQuant (Ours)** | **18.1** | **0.965** | **0.292** | **17.5** |
>
> ### Code
> As promised in the abstract, we will release the code and models upon publication. The code is available at this [anonymous link](https://anonymous.4open.science/r/svdquant-anonymous-81C0/).

---

> ### Author Response · Authors · 2024-11-23
> **Author Response to Reviewer R3NW (Continued)**
>
> ### Rank Ablation
> In our original submission, we have included an ablation study on rank selection in Section 5.2 (Trade-off of increasing rank) and Figure 10. We found that a rank of 32 effectively restores the image quality of the original 16-bit model with only about 5% latency and memory overhead. While increasing the rank to 64 further improves image quality, the overhead (~10%) becomes substantial. Additionally, Figure 5 shows that the first 32 singular values of the transformed weights are significantly larger, explaining why rank 32 is optimal.
>
> ### Robustness of Different Weight Distribution
> Our method's robustness to non-Gaussian distributions can be demonstrated both theoretically and empirically:
> * Theoretically, while Proposition 4.2 uses Gaussian assumptions for simplicity, our framework can be easily generalized to any distribution satisfying:
> $$\mathbb{E}[\max(|{\boldsymbol{R}}|)] \le c\cdot \mathbb{E}[||{\boldsymbol{R}}||_F],$$ where $c$ characterizes the distribution's skewness. The core relationship between matrix norm and quantization error holds across distributions, though the precise scaling factor may vary. See Proposition 2 and Appendix A.2 in our revised manuscript for details.
> * Empirically, QLoRA [1] shows that neural network weights typically follow near-normal distributions except for a small number of outliers (see its Appendix F). Our low-rank branch helps mitigate these outliers, leaving a more regular distribution that aligns better with our theoretical assumptions.
>
> ### Mixed Precision
> Our LoRunner engine supports mixed precision by decoupling the precision-dependant low-bit branch computation loop and the epilogue for low-rank branches. Adapting to any other precisions only requires updating the main loop, making LoRunner highly flexible to different precision configurations.
>
> ### Results below 4-Bit Quantization
> As requested by the reviewer, below we show the results of sub-4-bit quantization. Our SVDQuant can still generate images in the 3-bit settings on both PixArt-∑ and FLUX.1-schnell, performing much better than SmoothQuant [7]. Below this precision (e.g., W2A4 or W4A2), SVDQuant cannot produce images either, since 2-bit symmetric quantization is essentially ternary quantization. Prior work [9,10] has shown that ternary neural networks require quantization-aware training even for weight-only quantization to adapt the weights and activations to the low-bit distribution.
>
> | Model | Precision | FID (↓) | Image Reward (↑) | LPIPS (↓) | PSNR (↑) |
> |---|---|---|---|---|---|
> | PixArt-∑ | Our W4A4 | 20.1 | 0.90 | 0.394 | 16.2 |
> |  | SmoothQuant W3A4 | 401 | -2.25 | 0.784 | 7.83 |
> |  | Our W3A4 | 62.2 | 0.10 | 0.572 | 13.7 |
> |  | Our W3A3 | 176 | -1.38 | 0.669 | 12.5 |
> | FLUX.1-schnell | Our W4A4 | 18.1 | 0.965 | 0.292 | 17.5 |
> |  | SmoothQuant W3A4 | 22.2 | 0.857 | 0.529 | 13.8 |
> |  | Our W3A4 | 19.2 | 0.882 | 0.446 | 15.0 |
> |  | SmoothQuant W3A3 | 42.5 | 0.283 | 0.671 | 13.2 |
> |  | Our W3A3 | 30.4 | 0.677 | 0.559 | 14.0 |
>
> ### Speed Breakdown
> We compare the latency and memory usage of a single forward pass for INT4 FLUX.1 DiT on a 4090 desktop GPU in the below table. Without optimization, SVDQuant incurs about 18% latency overhead due to the low-rank branches. LoRunner significantly reduces this overhead, achieving a similar latency to naive INT4. Memory usage remains unaffected by LoRunner, with the low-rank branches adding only ~3% memory costs.
> | Device | Latency (ms) | Memory (GiB) |
> |---|---|---|
> | Naive INT4 | 212 | 6.3 |
> | SVDQuant (rank=32) | 250 | 6.5 |
> | SVDQuant + LoRunner | 218 | 6.5 |
>
> ### Typos
> Thanks for pointing them out. We have fixed them in our new revision.
>
> #### References
> 1. QLoRA: Efficient Finetuning of Quantized LLMs
> 2. ZeroQuant-V2: Exploring Post-training Quantization in LLMs from Comprehensive Study to Low Rank Compensation
> 3. QNCD: Quantization Noise Correction for Diffusion Models
> 4. Q-DiT: Accurate Post-Training Quantization for Diffusion Transformers
> 5. MixDQ: Memory-Efficient Few-Step Text-to-Image Diffusion Models with Metric-Decoupled Mixed Precision Quantization
> 6. EfficientDM: Efficient Quantization-Aware Fine-Tuning of Low-Bit Diffusion Models
> 7. SmoothQuant: Accurate and Efficient Post-Training Quantization for Large Language Models
> 8. QuaRot: Outlier-Free 4-Bit Inference in Rotated LLMs
> 9. BitNet: Scaling 1-bit Transformers for Large Language Models
> 10. The Era of 1-bit LLMs: All Large Language Models are in 1.58 Bits

---

> > ### Comment · Reviewer_R3Nw · 2024-11-24
> >
> > Thank you for your explanation and for providing the requested data. I increase my score.

---

### Author Response · Authors · 2024-11-23
**General Response to All Reviewers and ACs**

We sincerely appreciate all reviewers' efforts for the insightful and thoughtful comments. We are glad that the reviewer recognized the following strengths:
* **Novelty & Contribution**: Combining low-bit quantization and low-rank decomposition to mitigate quantization outliers is interesting (Reviewers 2HuL, 2VPm, 44PJ, and KPne). The authors have also implemented a novel inference engine that effectively minimizes the computational overhead (Reviewers ntPK and 2VPm).
* **Experiments**: The experiments are conducted on state-of-the-art diffusion models with strong and robust results (Reviewers 2HuL and ntPK).
* **Presentation**: The paper is well-written and easy to follow (Reviewers 2HuL, 2VPm, and 44PJ). The figures are well illustrated, and the math presentations are insightful (Reviewer KPne).

In addition to the pointwise responses below, we first clarify and highlight our contribution and novelty and then summarize the major changes in our revision:
* Novelty & contribution:
    1. Unlike prior methods that redistribute quantization difficulty (e.g., SmoothQuant [1] between weights and activations or SpinQuant [2]/QLLM [3] within channels), our core idea is to introduce a high-precision low-rank branch to **absorb outliers** in both weights and activations.
    2. We **co-designed an inference engine LoRunner with kernel fusion** to implement our quantization algorithm. Without it, the overhead of the low-rank branch will cancel out the quantization speedup, making the quantization algorithm impractical on modern GPUs.
* Major revisions:
    1. **Comparison with LLM baselines**: As requested by reviewer 2VPm, we additionally adapted LLM quantization methods (QLLM [3], QuaRot [4], AffineQuant [5] and SmoothQuant [1]) to diffusion models and evaluated them on both PixArt-∑ and FLUX.1-schnell (Apendix D.4 and Table 5). SVDQuant outperforms all these baselines, especially a wide margin on PixArt-∑ .
| Model | Method | FID (↓) | Image Rewad (↑) | LPIPS (↓) | PSNR (↑) |
|---|---|---|---|---|---|
| PixArt-∑ | Naive | 206 | -1.23 | 0.762 | 9.08 |
|  | SmoothQuant | 48.6 | 0.617 | 0.607 | 12.9 |
|  | QLLM | 35.8 | 0.763 | 0.581 | 13.1 |
|  | AffineQuant | 29.6 | 0.816 | 0.540 | 14.5 |
|  | QuaRot | 28.2 | 0.847 | 0.459 | 15.3 |
|  | **SVDQuant (Ours)** | **20.1** | **0.898** | **0.394** | **16.2** |
| FLUX.1-schnell | Naive | 18.1 | 0.962 | 0.345 | 16.3 |
|  | SmoothQuant | 18.4 | 0.943 | 0.323 | 16.7 |
|  | QLLM | 18.3 | 0.959 | 0.295 | 17.3 |
|  | AffineQuant | 22.8 | 0.937 | 0.292 | 16.9 |
|  | QuaRot | 19.3 | 0.951 | **0.287** | 17.4 |
|  | **SVDQuant (Ours)** | **18.1** | **0.965** | 0.292 | **17.5** |
    2. **Bitwidth tradeoffs**: As requested by reviewer 2VPm, we evaluated image quality across different bitwidths and highlighted tradeoffs for various quantization methods (Appendix D.5, Figure 18). SVDQuant consistently delivers better image quality across all bitwidths, with 4-bit results surpassing naive 6-bit quantization. We also discuss the results below 3 bits, as suggested by reviewers R3NW and 2HuL.
    3. **Pre-trained LoRA support**: We demonstrate that with LoRunner, SVDQuant can support off-the-shelf LoRA without the need for re-quantization (Section 3.2 Integrate with LoRA, Figure 9 and 17).
    4. **Skewed weight analysis**: As requested by reviewers R3Nw, we analyze the quantization error reduction of skewed weight distribution (Proposition 4.2 and Appendix A.2).
    5. **Related work discussion**: As requested by review R3Nw, we discussed the similarities and differences of related works in Table 2.
    6. **Additional latency results**: As requested by reviewers R3NW, ntPk and 44PJ, we compare the latency of a single forward step of FLUX with naive INT8 and INT4 models (Appendix D.3, Table 4). LoRunner achieves a 1.3× speedup over INT8 and reduces low-rank branch overhead to 3% compared to naive INT4.
    7. **Comparison with Q-Diffusion and QuEST**: As requested by reviewer 2HuL, we compared SVDQuant with Q-Diffusion [6] and QuEST [7] on Stable Diffusion 1.4. SVDQuant demonstrates significantly better LPIPS and PSNR.
    8. **Precise Wording**: Following reviewer KPne’s suggestion, we revised several statements for clarity and precision.

### References
1. SmoothQuant: Accurate and Efficient Post-Training Quantization for Large Language Models
2. SpinQuant: LLM quantization with learned rotations
3. QLLM: Accurate and Efficient Low-Bitwidth Quantization for Large Language Models
4. QuaRot: Outlier-Free 4-Bit Inference in Rotated LLMs
5. AffineQuant: Affine Transformation Quantization for Large Language Models
6. Q-Diffusion: Quantizing Diffusion Models
7. QuEST: Low-bit Diffusion Model Quantization via Efficient Selective Finetuning

---

### Meta-Review · Area_Chair_jNoN · 2024-12-19

**Metareview:**

This paper proposed a novel 4-bit post-quantization paradigm called SVDQuant for diffusion models. It is well written. The experimental parts, a variety of latest diffusion models were evaluated with about 3.5x speedup , demonstrating the universal effectiveness of the proposed method. The main weakness is that the novelty (or contribution) of the key idea per se is not that new, but the combinations of them are indeed really interesting and useful, and important. I think it is a strong submission.

**Additional Comments On Reviewer Discussion:**

1. Reviewer R3Nw's main concerns were the the level of contribution, and lack of theoretical analysis, while asking for providing ablation study of the rank,  more evidence of its robustness of different weight distribution, etc. In the rebuttal, the authors addressed most of the concerns though only providing some theoretical insight for the theoretical analysis, and Reviewer R3Nw raised the score.

2. Reviewer 2HuL's main concerns were lack of results on sub 4-bit quantization and comparisons with MixDQ and ViDiT-Q. In the rebuttal, the authors provided the required experiments which successfully satisfied the Reviewer 2HuL.

3. Reviewer ntPk's main concerns were lack of W8A8 baseline to the latency comparison, and lack of group quantization. In the rebuttal, the authors provided the required experiments which addressed the concerns.

4. Reviewer 2Vpm's main concerns were that the idea of splitting low-rank and low-bit branches is not good to overcome quantization errors: it could lead to 5~10% overheads in the paper (line 314) compared with QLLM. Moreover, Reviewer 2Vpm argued that the experiments are not very solid without including SpinQuant, AffineQuant. In the rebuttal, they authors applied RTN to both QuaRot and SVDQuant for fair comparisons. They further appled the GPTQ to both methods and show the results. Reviewer 2Vpm raised the score accordingly but no response for authors's further response. From my own reading, the rebuttal mostly addressed the reviewer's concern.

5. Reviewer 44PJ's main concerns were the choice of lambda (difficult to determine in practice) and lack of inference time results. Another concern is that the ideas not novel per se but its combination is interesting. The authors provide answers to most concerns but reviewer not responded. I think most concerns were addressed but I also agree that idea per se is not very novel but still its combination is very interesting and useful.

6.  Reviewer KPne's main concerns were lack of sufficient reasoning for some strong statements like "Weight-only quantization cannot accelerate diffusion models", "Weight-only quantization cannot accelerate diffusion models", as well as a lack of discussion of the computation flow. In the rebuttal, the authors revised their manuscript accordingly though reviewer did not give further response. I agree with reviewer's comments and also glad that the authors revised them.

---

### Decision · Program_Chairs · 2025-01-22

Accept (Spotlight)